# Circulating miRNAs as Biomarkers for Mitochondrial Neuro-Gastrointestinal Encephalomyopathy

**DOI:** 10.3390/ijms22073681

**Published:** 2021-04-01

**Authors:** Mark Mencias, Michelle Levene, Kevin Blighe, Bridget E. Bax

**Affiliations:** 1Molecular and Clinical Sciences, St. George’s, University of London, London SW17 0RE, UK; mmencias@sgul.ac.uk (M.M.); michelle.levene@hotmail.co.uk (M.L.); 2Clinical Bioinformatics Research Ltd., London W1B 3HH, UK; kevin@clinicalbioinformatics.co.uk

**Keywords:** MNGIE, Mitochondrial neurogastrointestinal encephalomyopathy, miRNA, biomarker, mitochondrial disease, thymidine phosphorylase, erythrocyte encapsulated thymidine phosphorylase, outcome measures, *miR-34a-5p*

## Abstract

Mitochondrial neurogastrointestinal encephalomyopathy (MNGIE) is an ultra-rare disease for which there are currently no validated outcome measures for assessing therapeutic intervention efficacy. The aim of this study was to identify a plasma and/or serum microRNA (miRNA) biomarker panel for MNGIE. Sixty-five patients and 65 age and sex matched healthy controls were recruited and assigned to one of four study phases: (i) discovery for sample size determination; (ii) candidate screening; (iii) candidate validation; and (iv) verifying the performance of the validated miRNA panel in four patients treated with erythrocyte-encapsulated thymidine phosphorylase (EE-TP), an enzyme replacement under development for MNGIE. Quantitative PCR (qPCR) was used to profile miRNAs in serum and/or plasma samples collected for the discovery, validation and performance phases, and next generation sequencing (NGS) analysis was applied to serum samples assigned to the candidate screening phase. Forty-one differentially expressed candidate miRNAs were identified in the sera of patients (*p* < 0.05, log_2_ fold change > 1). The validation cohort revealed that of those, 27 miRNAs were upregulated in plasma and three miRNAs were upregulated in sera (*p* < 0.05). Through binary logistic regression analyses, five plasma miRNAs (*miR-192-5p*, miR-193a-5p, *miR-194-5p*, *miR-215-5p* and *miR-34a-5p)* and three serum miRNAs (*miR-192-5p*, *miR-194-5p* and *miR-34a-5p*) were shown to robustly distinguish MNGIE from healthy controls. Reduced longitudinal miRNA expression of *miR-34a-5p* was observed in all four patients treated with EE-TP and coincided with biochemical and clinical improvements. We recommend the inclusion of the plasma exploratory miRNA biomarker panel in future clinical trials of investigational therapies for MNGIE; it may have prognostic value for assessing clinical status.

## 1. Introduction

Mitochondrial neurogastrointestinal encephalomyopathy (MNGIE, Online Mendelian inheritance in Man #603041, Genome Database accession #9835128) is an ultra-rare autosomal recessive disease caused by mutations in *TYMP*, the nuclear gene which encodes for the enzyme, thymidine phosphorylase. The resulting enzyme deficiency leads to plasma and tissue accumulations of thymidine and 2′-deoxyuridine, which create imbalances within the mitochondrial deoxynucleotide pools that, in turn, produce multiple mitochondrial deoxyribonucleic acid (mtDNA) deletions, depletion and site-specific point mutations, ultimately causing mitochondrial dysfunction [1,2,3,4,5]. MNGIE manifests clinically as severe gastrointestinal dysmotility, sensorimotor peripheral neuropathy, chronic progressive external ophthalmoplegia, severe muscle weakness and progressive leukoencephalopathy. The disease is relentlessly progressive, with patients usually dying from a combination of gastrointestinal and nutritional failure as well as neuro muscular disability at an average age of 37.6 years [6].

Although there is currently no universally established treatment to prevent or reverse the inexorable fatal clinical deterioration in patients with MNGIE, a number of experimental therapeutic approaches have been investigated over the past two decades [7,8,9,10,11,12,13,14,15,16,17,18,19]. Unique challenges face the translation of candidate therapeutics into approved treatments for rare diseases, and MNGIE is no exception. The small patient population and phenotypic heterogeneity of MNGIE, together with the small number of reported cases, an incomplete knowledge of the disease’s natural history and the absence of prospective clinical studies and validated outcome measures to quantitate disease progression have all contributed to the difficulty in selecting clinical trial endpoints for this disorder. Current clinical trial protocols for MNGIE have selected a range of primary and secondary clinical outcome measures that will enable the capture of clinical effect in a condition for which there is phenotypic variability (e.g., gastrointestinal and/or neuropathic), through the use of instruments that are not disease specific but have been shown to correlate significantly with both generic and disease targeted long-standing (legacy) instruments.

To expedite the development of drugs for rare diseases, the regulatory agencies have endorsed the need for flexibility in the review process and may consider approving a therapy based on a surrogate endpoint, such as a biomarker that is predictive of clinical benefit [20]. The definition of a biomarker is a biological characteristic that can be objectively measured and evaluated as a signal of normal biological or pathological processes and thus has the potential to improve diagnosis, predict disease manifestation and monitor responses to therapeutic intervention. Whereas clinical outcome measures directly assess how an individual feel, functions or survives, biomarkers substitute for a clinical endpoint and link a measurement to a prediction of the effect of the treatment on the clinical outcome of interest. Biomarker endpoints have the advantage of reducing drug approval time compared to the wait for the realisation of clinical endpoints, thereby expediating patient access to therapeutic benefits. Indeed, biomarker endpoints, both validated and unvalidated, have been used to support drug approvals through expedited pathways available for medicinal products intended to treat seriously debilitating or life-threatening diseases [21].

MicroRNAs (miRNAs) are a class of endogenous non-coding ribonucleic acids (RNAs) of 21–24 nucleotides in length that induce post-transcriptional gene silencing. In the majority of cases the miRNAs bind via complementary base-pairing with the 3′ untranslated region (3′ UTR) of target messenger RNAs (mRNAs) to induce mRNA degradation or translation repression [22]. miRNAs can be found in virtually in all cellular compartments and have been shown to exist in a stable state in most extracellular fluids through their binding with argonuate proteins and encapsulation by extracellular vesicles [23]. A growing body of evidence shows that miRNAs have unique expression profiles in the extracellular fluids of patients with a range of diseases, compared to healthy individuals, and thus may have clinical relevance as disease biomarkers [24,25,26].

The absence of validated outcome measures to quantitate MNGIE disease progression and determine the effectiveness of potential therapies led us to investigate whether a panel of miRNAs could serve as a potential biomarker for MNGIE. Our previously reported feasibility study showed that compared to age and sex matched healthy controls, there was significant dysregulation in a number of miRNAs in the sera of patients with MNGIE [27]. To further explore the clinical applicability of circulating miRNAs as biomarkers of MNGIE, we report here an expanded miRNA expression profiling study comprising four phases, with each phase recruiting different cohorts of patients and healthy controls.

## 2. Results

### 2.1. Study Design

The study was conducted in four phases according to the study design depicted in Figure 1. The aim of the discovery phase was to permit a power calculation for sample size determination for subsequent phases. In the second, candidate screening phase samples were analysed by next generation sequencing (NGS). The aim of this phase was to identify suitable candidate miRNAs and normalisers for verification by real-time quantitative polymerase chain reaction (RT-qPCR) in the validation phase. An additional aim of the validation phase was to determine whether the type of blood biofluid (plasma or serum) had an effect on the miRNA measurement. Finally, the responses of the validated miRNAs to the compassionate treatment of patients with erythrocyte-encapsulated thymidine phosphorylase (EE-TP), an enzyme replacement therapy under clinical development for MNGIE, were assessed in the performance phase of the study.

### 2.2. Study Population

Age and sex matched healthy controls were recruited into each study cohort. The discovery cohort consisted of five males and five females per participant group (healthy control and patient), with an average participant age of 31 ± 6.1 years (mean ± SD). To examine the possible effects of disease progression on the miRNA expression profile, the candidate screening cohort was subdivided into three age range groups: ≤19 years (3 males, 2 females; mean age 13 ± 4.2 years), 20–29 years (3 males, 5 females; mean age 23.3 ± 2.6 years) and ≥30 years (4 males, 3 females; mean age 33.9 ± 3.2 years). The validation cohort consisted of both serum and plasma samples to ascertain whether the two biofluids provided similar miRNA profiles. The participant demographics for the serum samples were: ≤19 years (4 males, 2 females; mean age 11.3 ± 4.6 years), 20–29 years (3 males, 2 females; mean age 22.6 ± 0.6 years) and ≥ 30 years (5 males, 1 female; mean age 39.0 ± 11.1 years); and for the plasma samples: ≤19 years (0 male, 4 females; mean age 17.0 ± 1.4 years), 20–29 years (4 males, 5 females; mean age 24.4 ± 2.6 years) and ≥30 years (3 males, 2 females; mean age 36.2 ± 4.4 years). The performance cohort consisted of serum samples which had been collected longitudinally from four patients with MNGIE who had received treatment with EE-TP.

### 2.3. Discovery Phase

#### 2.3.1. Data Quality Control (QC)

Box and whisker plots comparing distributions revealed a normalised dataset with no identified sample outliers; see Figure A1. A comparison of sample distributions on a pairwise basis, with different regression fits, revealed no outliers. Some lower tails in distributions were observed in some pairwise comparisons via locally estimated scatterplot smoothing (LOESS, see Section 4.5.1), but this was most likely due to biological variability (data- not shown).

Unsupervised clustering analysis revealed a natural separation between healthy control and patient serum samples, and showed that the miRNA profile amongst the control group was similar, whereas it exhibited greater variability in MNGIE samples, based on pairwise Euclidean distances; see Figure 2. This was also evident via principal component analysis (PCA), where variability in controls was mostly confined to principal component 1 (PC1) (15.66% explained variation), whilst variability in MNGIE occurred along both PC1 and PC2 (9.37% variation). No PC segregated MNGIE from controls in a conclusive fashion. Of the PCs accounting for large (>5%) amounts of variation, PC2 appeared to segregate both groups along an axis at, roughly, eigenvalue 0.

#### 2.3.2. Differentially Expressed miRNAs

A total of 27 miRNAs were differentially expressed between healthy control and MNGIE serum samples when applying the threshold of *p* ≤0.05 (Benjamini–Hochberg adjusted) and an absolute fold-change > ± 2; see Table 1. These included 14 upregulated and 13 downregulated miRNAs; see Figure 3.

Supervised clustering of these differentially expressed miRNAs was able to segregate healthy controls from MNGIE; see Figure 4. Based on the heatmap shading, the majority of these miRNAs were either exclusively highly expressed in healthy control serum or MNGIE disease serum. A few miRNAs, in addition to being highly expressed in controls, were also highly expressed in 1 or 2 MNGIE samples (e.g., *miR-409−3p*). These were assumed genuine biological effects in which the MNGIE patient sample in question may be at a different stage of disease. This data strongly suggests that, serum miRNAs profiles can segregate the healthy and disease groups.

A power analysis of the data at a 0.95 level of confidence and for a *p*-value of 0.05 revealed that 17 samples per group would be required to detect an effect in a follow-on validation study.

### 2.4. Candidate Screening Phase

#### 2.4.1. Descriptive Analysis of Sequencing Data

One patient sample (SP1) yielded a very low library, rendering it unfit for pooling with the other libraries and was therefore removed from the downstream sequencing work. All other samples passed the quality control criteria (see Table A1). Expression levels of *miR-103a-3p*, *miR-191-5p*, *miR-451a*, *miR-23a-3p* and *miR-30c-5p* were within the expected range for miRNAs in biofluids. The dCp (*miR23a*-*miR451a*) values were 5 or less, indicating no haemolysis (Figure A2).

The QC spike-ins added during the RNA isolation procedure showed count correlations (R2) between the samples in the range of 0.95–0.99, therefore demonstrating the reproducibility of the NGS data. miRNA sequencing generated 745 million raw sequencing reads with an average of 19.1 million raw reads/sample. Following Unique Molecular Index (UMI) correction, an average of 1.55 million reads were obtained per sample. Each sample consisted of reads that were classified into the following categories: miRNA, small RNA, predicted putative miRNA, genome-mapped, outmapped or high abundance (e.g., ribosomal RNA, polyA, polyC, mitochondrial RNA) and unmapped (reads which did not align to the reference genome); see Table A2. Averaging the data from all 39 samples, 59.7% of the UMI-corrected reads were mappable; 26.4% were mapped to MiRBase20, 18.1% a small RNA database, and 12.6% were aligned to the GRCh37 reference genome, but not to miRNA or small RNAs. In additional, 0.04% of the reads represent putative miRNAs that mapped to the reference genome outside the miRNA locus and were predicted using mirPara64 and miRbase; see Table A2. The Phred quality scores for all UMI-corrected reads, both control and patient samples, were above 30% (>99.9% accuracy), representing a high and uniform quality of reads. This refers to the mean Phred scores per read (per sequence quality) and to Phred scores along all reads (per base quality); see Figure A3 and Figure A4, respectively. Read length distribution after filtering of the adapters is shown in Figure A5. The majority of reads were 18–23 nucleotide bases in length, which corresponds to the expected length of mature miRNAs. This data demonstrates a high technical performance of miRNA sequencing and a good sample quality.

#### 2.4.2. miRNA Expression

Following data mapping and counting to relevant entries in mirbase20 which contains 2578 mature miRNA sequences, a total of 1219 miRNAs were identified in the 39 libraries. By employing tags per million mapped reads (TPM) as a unit of expression, 170 and 121 of known miRNAs were expressed at ≥1TPM and ≥10TPM, respectively, in each of the 39 serum samples, and these were accepted for further analysis; see Figure A6. Due to the discovery study revealing a greater variability in the miRNA profiles in serum samples from MNGIE patients, compared to healthy controls, it was hypothesised that this was related to the extent of disease progression. To investigate this further, we examined the expression profiles of miRNA in samples collected from three different age ranges: ≤19, 20–29 and ≥30 years of age; these broadly corresponding to early, mid and late disease progression status, respectively. To explore biological differences between these sample groups, a PCA was firstly conducted on the top 50 miRNAs that showed the largest variation across all samples. Figure 5 shows a clear division in the miRNA expression profiles between the MNGIE and healthy control groups along PC2, but particularly in the age group ranges 20–29 and ≥30 years, thereby demonstrating the similarity of samples within the study groups.

By examining the component loading it was shown that *miR-122-5P* separates healthy from disease samples and was therefore the primary driver of this segregation; see Figure 6.

#### 2.4.3. Differentially Expressed miRNA Profile

Of the 1219 known miRNAs identified in this screening study, a total of 80 miRNAs were differentially expressed: 55 of these were upregulated in MNGIE compared to healthy controls, and 25 were downregulated; see Table 2.

Supervised hierarchical clustering of the data showed that in general, the significantly differentially expressed miRNAs are able to distinguish between MNGIE disease and healthy control samples; see Figure 7.

To determine the most meaningful miRNAs in terms of predicting disease status and disease stage, a multinomial elastic-net penalised regression model was constructed using the 80 differentially expressed miRNAs as predictors and age range groups as outcome. This resulted in a refined signature of 41 miRNAs with an area under curve (AUC) of 0.803 by receiver operating characteristic (ROC) analysis; see Figure 8. Of these miRNAs, *miR-122-5p*, *miR-192-5p*, *miR-206* and *miR-34a-5p* were shown to increase in expression across age groups and with higher expression levels in MNGIE, compared to all healthy control age range groups. In addition, *miR-362-5p* revealed a decreased expression with age in patients with MNGIE, compared to healthy controls.

#### 2.4.4. NormFinder Analysis

One of the objectives of the candidate screening study was to identify candidate miRNAs that would be suitable to use as normalisers in the downstream qPCR validation study. The Normfinder algorithm was used to identify suitable reference miRNAs, and Table 3 shows the top ten miRNAs that were most stably expressed across all samples. On the basis of those results, *miR-30e-5p*, *miR-425-5p*, *let-7i-5p* and *let-7b-5p* were selected as potential normalisers for the qPCR validation phase.

### 2.5. Validation Phase

The two aims of this study phase were firstly to determine whether the method of blood sample preparation (serum or plasma) affected the measurement of miRNAs and secondly to verify the findings of the candidate screening study and finalise a miRNA panel. The 80 significantly differentially expressed miRNAs; the four contender normalisers that were identified in the candidate screening study; and a few other differentially expressed miRNAs of interest were selected for qPCR analysis.

#### 2.5.1. Extraction Efficiency and Sample QC

The efficiency of RNA extraction was monitored by employing a subset of synthetic miRNAs that simulate high (UniSp2) and medium (UniSp4) expression signals. With the exception of plasma samples from three patients (PP8, PP11 and PP12) and two healthy controls (PC5 and PC 13), which showed higher raw quantification cycle (Cq) values, the steady expression levels of these spike-ins indicate that the extraction efficiency was similar for the majority of samples (Figure A7). Complementary DNA (cDNA) synthesis and qPCR efficiency were controlled by the inclusion of spike-ins UniSp6 and UniSp3, respectively. These miRNA spike-ins were shown to be expressed at a consistent level in all samples, including the blank assay, thereby indicating that the reverse transcription and qPCR were successful and none of the samples contained inhibitors (Figure A7).

Sample mRNA signals were examined by the measurement of *miR-103*, *miR-30c*, and *miR-23a* which are expressed at a consistent level in a majority of sample types. Higher Cq values were seen in serum samples collected from four healthy controls (SC37, SC38, SC39 and SC41), and three plasma samples (patient: PP1 and PP12; healthy control: PC5) indicating lower miRNA content in these samples. Interestingly, the four healthy control plasma samples affected were all collected from individuals who were aged ≤13 (Figure A7). *miR-142-3p*, which is highly expressed in platelets, was also measured to assess sample contamination by this cell type. The results showed a steady level of expression across all samples (Figure A7). Sample contamination by miRNAs released as a result of sample haemolysis was assessed from the dCq value of (*miR-23a*–*miR-451a*). All samples had a value less than 7, indicating that the collected samples were not affected by erythrocyte miRNA contamination (Figure A7). No samples were excluded from the final analysis.

#### 2.5.2. Determination of the Most Stable Reference Genes

Of the 95 miRNAs present on the miRNA PCR panel, 15 assays were detected in all serum and plasma samples, with an average detection of 64 assays per sample; see Figure A8.

Using NormFinder, the stabilities of the global miRNA profiles detected in all plasma and serum samples were examined and the top ten most candidate reference miRNAs were identified. *miR-30e-5p* and *miR-15b-5p* provided the lowest stability values of 0.00460 and 0.00577, respectively, and when combined provided a stability score of 0.00347. Thus, the average Cq values of these two miRNAs for each sample were employed as the normalisers. To evaluate the extent to which normalisation reduced data dispersion across each sample, box and whisker plots were employed; with the exception of one disease sample and one healthy control sample, these evaluations generally demonstrated a consistent data distribution; see Figure A9.

#### 2.5.3. Identification of Differentially Expressed miRNA in MNGIE Plasma and Serum Samples

A comparison between the MNGIE and healthy groups using the students two-tailed t-test revealed 44 differentially expressed miRNAs in plasma and nine differentially expressed miRNAs in serum. Following adjustment for false discovery rate using a Benjamini–Hochberg correction at a significance level of *p* < 0.05 and filtering data with a Log_2_ fold change > 1 (upregulated) and < −1 (downregulated), the data sets were reduced to 33 plasma miRNAs of which 27 were upregulated and 6 downregulated, and three serum miRNAs were upregulated; see Figure 9 and Table 4. Seventeen of these miRNAs are common to the refined serum miRNA signature established following the construction of a multinomial elastic-net penalised regression model of the sequencing data.

#### 2.5.4. Meta-Analysis of qPCR and Sequencing Data

In order to corroborate results across the candidate screening and validation phases, a meta-analysis of the qPCR and NGS data was performed. A meta-analysed combined *p*-value was calculated for each miRNA for serum miRNA-seq/plasma qPCR and serum miRNA-seq/serum qPCR and the *p*-values ranked; see Table 5. A combined *p*-value from the meta-analysis of MNGIE versus control was set at a threshold of < 0.0001, resulting in a panel of five plasma miRNAs (*miR-192-5p*, *miR-193a-5p*, *miR-194-5p*, *miR-215-5p* and *miR-34a-5p*) and a panel of three serum miRNAs (*miR-192-5p*, *miR-194-5p* and *miR-34a-5p*).

#### 2.5.5. Elucidation of MNGIE miRNA Panel

In order to define a final MNGIE disease miRNA panel for the overall study, binary logistic regression analyses were conducted by regressing each miRNA independently to the main end-point (disease versus healthy). The results from these analyses demonstrated that the miRNAs identified from the meta-analyses still retained a high level of statistical significance (Table A3).

To evaluate whether these five miRNAs could serve as potential biomarkers for MNGIE disease, ROC curve analyses were performed independently on each of the regression models constructed. ROC analysis demonstrates the trade-off between sensitivity and specificity and a good biomarker should display both high sensitivity and high specificity. ROC curves for each miRNA in plasma and serum are shown in Figure 10. The AUC quantifies the biomarker potential for each candidate miRNA, where the higher the value, the better the candidate miRNA is in distinguishing MNGIE disease from healthy controls. Each of the five plasma miRNAs, *miR-34a-5p*, *miR-193a-5p*, *miR-215-5p*, *miR-192-5p* and *miR-194-5p* yielded AUC values of 0.977 (95% confidence interval (CI), 0.938–1, *p*= 0.019), 0.941 (95% CI, 0.867–1, *p* = 0.004), 0.908 (95% CI, 0.813–1, *p* =0.002), 0.850 (95% CI, 0.711–0.988, *p* = 0.005) and 0.804 (95% CI, 0.642–0.966, *p* = 0.009), respectively (Figure 10). The combined model for the five plasma miRNAs results in a perfect prediction with an AUC= 1. The three serum miRNAs, *miR-34a-5p*, *miR-192a-5p* and *miR-194-5p* had AUC values of 0.908, (95% CI, 0.813–1, *p* = 0.010), 0.824 (95% CI, 0.672–0.975, *p* = 0.005) and 0.803 (95% CI, 0.642–0.963, *p* = 0.013), respectively (Figure 10). The combined model for the three serum miRNAs provides an AUC of 0.851 (CI, 0.703–0.999). These results therefore reveal the probability of these miRNAs as biomarkers of MNGIE.

### 2.6. Performance Phase

The clinical utility of the validated miRNA panel was assessed by examining the expression fold change of the miRNA panel in the plasma of four patients, A, B, C and D who received up to 2, 5, 14 and 62 infusions of EE-TP respectively (Figure 11). The largest responses in terms of a sustained reduced expression, were noted in patient D, who also showed the greatest clinical and biochemical responses to EE-TP therapy. Clinical responses included improved MRC sum and sensory sum scores, improved distal sensation and weight gain. Biochemical improvements included metabolic correction of plasma thymidine and 2′-deoxyuridine concentrations and normalisation of creatine kinase activities (Table 6). Smaller decreases in the expression of all miRNAs were recorded for patients B and C at 3 and 4 months of therapy, respectively. Both patients demonstrated reductions in the plasma metabolites and patient C gained 2.9 kg in weight and reported a reduction in nausea and vomiting (Table 6). The subsequent increase in the miRNAs in patient C at 11 months of therapy coincided with intestinal bacterial overgrowth and the commencement of totalparenteral nutrition (TPN) for weight loss. Patient A demonstrated a decrease in the expression of *miR-34a-5p* only. Clinical improvements in this patient included a greater appetite, tingling sensations in the feet compared to no sensation pre-therapy and improved swallowing and dysgeusia. Biochemically, there were reductions in both thymidine and 2′-deoxyuridine.

Clinical observations and plasma metabolite concentrations during EE-TP therapy and at the time of miRNA assessment are reported in Table 6.

### 2.7. Target Gene Prediction and Enrichment Analysis

To annotate and speculate on the role of the five differentially expressed miRNAs, target genes of each individual miRNA were predicted using four databases. To be included in further enrichment analyses a gene target was required to be predicted in 2 of the 4 databases. A total of 536 downregulated genes were identified which were then subjected to Gene Ontology (GO) function analysis. Applying a threshold *p*-value < 0.05 revealed the enriched GO terms shown in Figure 12. In the molecular function associated category, the differentially expressed miRNAs were significantly associated with the following three terms: cytoskeletal protein binding (GO:0008092), calcium ion binding (GO:0005509) and molecular function regulator (GO:0098772). GO analysis of the biological processes-associated category showed that differentially expressed miRNAs were most significantly associated with the following terms: cell adhesion (GO:0007155), biological adhesion (GO:0022610), cell-cell adhesion (GO:0098609), cell–cell via plasma-membrane adhesion molecules (GO:0098742) and homophilic cell adhesion via plasma membrane adhesion molecules (GO:0007156). The most significant GO cell component associated terms of differentially expressed miRNAs included the following: plasma membrane part (GO:0044459), integral component of membrane (GO:0016021), intrinsic component of membrane (GO:0031224), integral component of plasma membrane (GO:0005887), intrinsic component of plasma membrane (GO:0031226), plasma membrane, (GO:0005886) membrane part (GO:0044425), cell periphery (GO:0071944) and neuron projection terminus (GO:0044306).

Kyoto Encyclopedia of Genes and Genomes (KEGG) pathway enrichment analysis showed that differentially expressed genes in the miRNA-mRNA regulatory network were enriched in six pathways: the notch signalling pathway (KEGG:04330), adherens junction (KEGG:04520), p53 signalling pathway (KEG: 04115), pancreatic cancer (KEG: 05212), glycosaminoglycan biosynthesis—heparan sulphate/heparin (KEGG: 00534) and N-glycan biosynthesis (KEGG: 00510); see Table 7.

A network of the five differentially expressed miRNAs and predicted genes targeted by more than two of the miRNAs was constructed (Figure 13). The results show that no gene was targeted by than more than three miRNAs; two genes (*CLIP3* and *GPR22*) were the target of three of the differentially expressed miRNAs and 52 genes were targeted by two of the miRNAs.

## 3. Discussion

Clinical outcome measures are employed as a means to accurately assess the efficacy of a treatment in terms of benefit (improved health) or risk (e.g., adverse reactions, hospitalisations, and death) and therefore selecting the most meaningful measure is crucial for the design of a valid clinical trial. Indisputably, many potentially effective drugs have failed to demonstrate efficacy due to the selection of inappropriate outcome measures. The development of outcome measures for rare diseases is however, particularly challenging given disease phenotypic heterogeneity, variable time frames for disease progression and incomplete knowledge of the disease pathophysiology. Regulators acknowledge the need for accelerating the rare disease drug development pathway and may approve a biomarker endpoint measure in the absence of validated endpoint measures for the intended patient population. Examples of rare disease drugs that have received approval based on biomarker primary endpoints include tiopronin for cystinuria (biomarker: reduction in urinary excretion of cystine) sapropterin dihydrochloride for phenylketonuria (reduction in plasma phenylalanine), agalsidase beta for Fabry disease (reduction of globotriaosylceramide storage granules in biopsied kidney interstitial capillaries), imiglucerase for Gaucher disease type I (composite of biomarker endpoints, including haemoglobin and platelet count) and sebelipase alfa for lysosomal acid lipase deficiency (normalisation of serum alanine aminotransferase levels) [28,29,30,31,32].

MNGIE is a complex multisystem disorder where patients present with a combination of cachexia, gastrointestinal dysfunction, and neuromuscular dysfunction; currently there are no validated clinical outcome measures which provide objective assessments of the clinical status of the patient. The identification of a biomarker for MNGIE would very likely lead to improved patient outcomes through accelerating patient access to new therapies, enabling early intervention and assessment of treatment efficacy. One of the recommendations of the recently published output from International Consensus Conference on MNGIE was the application of multi-omics analyses for the purpose of identifying biomarkers that fingerprint the main clinical features of the disease [33].

The discovery that miRNAs are detectable in extracellular biofluids has raised much interest in their potential as early non-invasive biomarkers for diseases. Studies reporting alterations in miRNA expression profiles in patients with muscular and neurological disorders led us to conduct a feasibility study in patients with MNGIE [27,34,35]. In this study we reported a significant dysregulation of 82 miRNAs in the serum of patients compared to age and sex matched healthy controls, however, a critical limitation of this study was the small number of patients recruited (*n* =5). Previous to this feasibility study, Yong et al. reported a miRNA profiling study in a single patient with MNGIE and a cohort of heterozygous family members, using a microarray-based screening [36].

To investigate the clinical applicability of circulating miRNAs as biomarkers of MNGIE with improved experimental rigour, we conducted an expanded expression miRNA profiling study whereby we recruited three further patient cohorts through an international collaborative effort. The workflow employed a rigorous four-phase methodological study design, incorporating a power analysis, sample and data quality control steps, spike-in controls, normalisers and candidate validation using RT-qPCR. For the candidate screening phase, NGS was employed for miRNA expression analysis, a state-of-the art technology which is suited for unbiased genome-wide miRNA expression profiling. For the discovery, candidate validation and performance phases, RT-qPCR was used, the gold standard method for expression profiling, having a high specificity and linear dynamic range of quantification. Since there are no standard reference or housekeeping miRNAs, potential normalisers were identified in the candidate screening stage. These were included in the custom miRNA qPCR panel employed in the validation phase where the stabilities of the global miRNA profiles detected in all samples were examined using NormFinder.

From the discovery phase it was noted that 10 of the 14 upregulated miRNAs and 7 of the downregulated miRNAs were previously identified in our published feasibility study [27]. Although unsupervised clustering analysis of the discovery data revealed a natural separation between healthy control and patient serum samples, and showed that the miRNA profile within the healthy control group was similar, it exhibited a greater variability in MNGIE samples, based on pairwise Euclidean distances. MNGIE is a progressive disorder, with an average diagnosis age of 18 years and a mean death age of 37.6 years, and thus we hypothesised that the variability noted in the MNGIE samples could be attributed to the stage of the disease affecting the miRNA expression profile. Thus, for the candidate screening phase we included an expression analysis of three patient sub-groups: ≤19, 20–29 and ≥30 years of age, these broadly representing early, middle and late disease stages. As a whole, the genome-wide miRNA expression profiling study of the candidate screening phase, revealed 80 candidate miRNAs with discriminative potential between patients with MNGIE and healthy controls; 55 of these were upregulated and 25 were downregulated in MNGIE. Through the construction of a multinomial elastic-net penalised regression model whereby the 80 differentially expressed miRNAs were predictors and age range groups were outcomes, the signature was refined to 41 miRNAs. Of these, *miR-122-5p*, *miR-192-5p*, and *miR-34a-5p* were shown to increase in expression across age groups and with higher expression levels in patients aged ≤ 19, compared to all healthy control age range groups. One of the typical features of full-blown MNGIE is the hepatic steatosis and cirrhosis, either disease related or induced by the long-term use of TPN. Interestingly, all three miRNAs have previously been shown in a number of studies to be associated with liver pathology, and consequently have been proposed as useful diagnostic biomarkers of liver injury [37,38,39,40]. The overexpression pattern of these miRNAs across the three disease age groups may therefore be explained by progressing liver pathology. The prognostic value of these miRNAs in predicting disease severity will require longitudinal validation using a uniform methodical study design; a limitation of this study is that the extent of disease progression was not scored in the patients recruited. Of note, the 20–29 age group patient group demonstrated overexpression of the haemolysis-related miRNAs (*miR-451a*, *miR-16-5p*, *miR-486-5p*, *miR-15a-5p and miR-15b-5p*) compared to healthy controls [41]. Although results from the quality control analysis demonstrated the dCp(*miR23a*-*miR451a*) levels were higher in the patient samples (range 0.2-2.5, samples SP 3, 4, 11, 14, 24, 25 and 26) compared to healthy control samples (range 0.9 to 1.1, samples SC 3, 4, 11, 14, 24, 25 and 26), these were well below the value that would normally indicate haemolysis. A more comprehensive understanding of the natural history study of MNGIE is also urgently required to enable disease progression quantification, identify outcome measures and assess the efficacy of investigational therapies; exploratory novel biomarkers such as miRNAs should be included in such studies.

As part of the validation phase, we aimed to determine whether the choice of blood biofluid had an effect on the miRNA profile. Numerous studies report the use of both blood fractions for identifying miRNA biomarkers in different pathological conditions. However, serum and plasma are often treated as similar or equivalent matrices and publications often fail to report the justification for their selection. Whereas plasma is the fluid that remains when blood coagulation is prevented through the addition of an anticoagulants, serum is the fluid that remains after allowing the blood to coagulate and doesn’t contain fibrinogens. The different preparation procedures have the potential to impact on miRNA recovery, making it difficult to compare data between studies using different matrices. In the current study, plasma and sera were collected from the same patients and healthy controls and processed downstream using identical procedures to determine whether there were any substantial differences in miRNA expression profiles between the two biofluids. A comparison between MNGIE and healthy control groups revealed 33 plasma miRNAs of which 27 were upregulated and 6 downregulated, and 3 upregulated serum miRNAs. Thus, this investigation confirms that the type of biofluid is a biological variable that has a large impact on the miRNA profile; compared to serum, plasma demonstrated a better detection of miRNAs when using qPCR, and the use of this biofluid in future studies is highly recommended. Potentially critical variables that contribute to these differences are the extent to which samples are subjected to haemolysis or platelet activation during processing, which could lead to the release of miRNAs and ribonucleases. Samples in this study were assessed for miRNAs which are released during these processes and were shown not to be over-expressed. The reason for the differences in miRNA expression between the two biofluids in this study is not known, but other considerations are the adsorption of miRNAs within the blood clot or the separator gel, leading to an artefactual depletion of miRNAs in serum. This investigation emphasises the importance of ascertaining the optimal biofluid fraction to employ and is an essential determinant if comparisons are to be made between different studies and thus ultimately advance miRNA biomarker discovery. Of note, 17 of the miRNAs detected by qPCR were common to the refined serum miRNA signature obtained following the construction of a multinomial elastic-net penalised regression model of the sequencing data.

To comprehensively assess the accuracy of circulating miRNAs biomarkers of MNGIE we performed a meta-analysis of the qPCR and sequencing data; this resulted in a panel of five significantly upregulated plasma miRNAs (*miR-192-5p*, *miR-193a-5p*, *miR-194-5p*, *miR-215-5p* and *miR-34a-5p*) and a panel of three significantly upregulated serum miRNAs (*miR-192-5p*, *miR-194-5p*, and *miR-34a-5p*) which were common to the plasma panel. Each miRNA retained a high level of statistical significance following binary regression analyses and showed considerable diagnostic power from the AUC values of ROC analysis. Furthermore, the recombination of the plasma miRNA panel exhibited a perfect prediction of AUC=1. The single best predictor of MNGIE is *miR-34a-5p,* having an AUC of 0.977 and was shown to be elevated across each age sub-group of disease, when compared to any healthy group. A decrease in expression of *miR-34a-5p* was noted during treatment with EE-TP in all four patients. Clinical responses recorded at the same time as sampling for miRNA analysis were generally associated with gastrointestinal features, including weight gain, improved appetite, swallowing and dysgeusia. An increase in expression of the miRNA panel was noted in patient C at 11 months and this coincided with intestinal bacterial overgrowth and commencement of TPN for weight loss in the preceding 4 months. Although deceases in plasma thymidine and 2′-deoxyuridine were observed in all four patients, only patient D was able to attain (and sustain) total metabolic correction, and intriguingly also sustained a decreased miRNA expression profile and an improvement in the peripheral neuropathy. The performance phase of this study was obviously limited by the number of patients recruited and absence of longitudinal sampling in two of the patients. To robustly test the performance of this miRNA panel and correlate miRNA expression profiles with clinical outcome data, future investigations would benefit from operating in tandem with a clinical trial of a therapeutic intervention, thereby exploiting the methodological rigor of a regulatory approved study design.

Theoretical insight into the underlying pathomolecular mechanisms of MNGIE was accomplished by conducting gene target prediction and enrichment and pathway analyses. The five upregulated plasma miRNAs, *miR-192-5p*, *miR-193a-5p*, *miR-194-5p*, *miR-215-5p* and *miR-34a-5p*, were selected and their putative gene targets and pathways in which the putative genes are involved were identified. KEGG pathway analysis (Table 7) revealed that the most significant pathways were all targets of *miR-34a-5p* and included Notch signalling, adherens junction and p53 signalling pathway. Notch signalling is a highly evolutionarily conserved pathway that regulates the balance between cell proliferation, differentiation and apoptosis. Relevant previous studies associating *miR-34a* with the Notch signalling pathway include that of Wang et al. who observed high expression levels *of miR-34a*, a low level of Notch signalling and neuronal apoptosis in the hippocampal neuronal spontaneous recurrent epileptiform discharges model [42]. In another study, Fan and co-workers studying age related apoptosis in the human lens epithelial cell, demonstrated that an overexpression of *miR-34a* and the subsequent inhibition of its target, Notch 2, induced mitochondria-mediated apoptosis through permeabilization of the mitochondrial outer membrane, cytochrome c release and activation of caspase-9 [43]. Thus, it could be speculated that the pro-apoptotic activity of *miR-34a* contributes to the degenerative pathology of MNGIE via mitochondria-mediated apoptosis.

Adherens junctions are plasma-membrane structures that mediate cell–cell adhesion by linking the plasma membrane to the actin cytoskeleton and are critical for the coordination of cell polarity, differentiation and signalling. The core molecular components of the junctions are the nectin–afadin and cadherin–catenin (E-cadherin, β-catenin, α-catenin, p120-catenin) complexes. A majority of evidence supporting the involvement of *miR-34a* in adherens junction signalling appears to have very little relevance to the MNGIE phenotype, reporting an inhibitory effect of *miR-34a* on epithelial-mesenchymal transition and cancer progression [44,45,46,47]. However, one study that has some commonality with MNGIE is that of Bukeirat et al. who demonstrated an overexpression of *miR-34a* leading to an increased blood brain barrier permeability and the disruption of the tight junction proteins, Zonula occludens, in cerebrovascular endothelial cells [48]. One of the clinical hallmark features of MNGIE is the leukoencephalopathy, a result of blood-brain barrier dysfunction leading to vasogenic oedema [49]. Although the bioinformatics analysis indicated that tight junction related genes were not targets of *miR-34a*, it is now recognised that adherens junctions and tight junctions are physically linked by the zonula occludens proteins, and thus this raises the possibly that disruption of the Zonula occludens was mediated via the inhibitory effects of *miR-34a* on adherens junction genes [50].

The p53 signalling pathway plays a salient role in co-ordinating the cellular response to different types of intrinsic and extrinsic stress signals, such as DNA damage, oncogene activation and hypoxia. In response to a stress signal, the nuclear transcription factor p53 is activated leading to the transcriptional regulation of the appropriate target genes to induce cell cycle arrest and/or apoptosis. A number of studies have confirmed that miRNAs play pivotal roles in the p53 pathway; whereas miRNAs have been reported to regulate the activity and function of p53 through direct repression of p53 or its regulators, p53 has also been shown to induce the transcription expression of a number of miRNAs, to promote the p53 response [51]. *miR-192*, *miR-194* and *miR-215*, three of the upregulated miRNAs identified in our study, have been reported to target *MDM2* and directly repress the expression of MDM2, a key regulator of p53, and thereby activate p53 [51,52,53]. In addition, *miR-34a* which was also found to be dysregulated in our study, has been reported to regulate p53 function through targeting a number of p53 negative regulators, including *MDM4* and *SIRT1* [52,53,54]. Other confirmed targets of *miR-34a* include *Bcl-2*, *E2F3*, *CDK4/6*, and *c-Myc*, thereby identifying *miR-34a* as a promoter of cell cycle arrest and apoptosis [55,56,57,58]. Interestingly, *miR-192*, *miR-194*, *miR-215* and *miR-34a* are also transcriptionally regulated by p53, therefore creating positive feedback loops with p53. Recent studies suggest that p53 may have a role in mtDNA homeostasis through its ability to translate to the mitochondrial matrix and interact with mtDNA polymerase γ in response to mtDNA damage induced by exogenous and endogenous insults [59]. In their study Achanta et al. demonstrated that an overexpression of p53 had a negative effect on the normal mitochondrial homeostasis, leading to a decrease in mtDNA abundance [60]. mtDNA depletion is one of the characteristic molecular findings in MNGIE, and it is therefore tempting to speculate that a miRNA-induced overexpression of p53 may contribute to the process of mtDNA loss. Experimental investigation is required to substantiate the relevance of the miRNA-p53 network in MNGIE and specifically, determine whether such interaction represents a homeostatic mechanism through the elimination of cells with dysfunctional mitochondria, or conversely, contributes to the disease pathology by amplifying cell damage.

One intriguing report that may be of relevance to the gastrointestinal pathology of MNGIE is the p53-induced expression of *miR-34* leading to the repression of c-kit expression in colorectal cancer cells via a conserved seed-matching sequence in the *c-Kit* 3′-UTR [61]. It is established that c-Kit is expressed in the interstitial cells of Cajal, the pacemaker cells located in the smooth muscle layers of the gastrointestinal tract, and which respond to enteric motor neurotransmitters [62]. The corresponding ligand of c-Kit is stem cell factor (SCF), which is synthesised by the smooth muscle cells of the gastrointestinal tract. The SCF/c-Kit signalling pathway is essential for normal development, maturation, and survival of interstitial cells of Cajal, and is required for maintaining their phenotype and functional networks. The inhibition of and loss-of-function mutations in *c-Kit* has shown to be associated with the trans-differentiation of interstitial cells of Cajal to a smooth muscle-like phenotype and gastrointestinal dysmotility disorders [63,64,65]. MNGIE is commonly associated with chronic intestinal pseudo-obstruction leading to severe gut motility failure. Examination of the small intestine neuromuscular pathology in patients with MNGIE has shown an absence of c-Kit-positive interstitial cells of Cajal around the myenteric plexus, intermuscular septa and within muscular plexus, which may be associated with cell death or trans-differentiation into a smooth muscle phenotype [66,67]. The plausible contribution of *miR-34a* to the pathomolecular mechanisms that underly the gastrointestinal aspects of MNGIE deserves further investigation.

Due to base-pair complementarity, a single miRNA is able bind to and regulate many mRNAs, and one mRNA can be regulated by multiple miRNAs. To visualise the miRNA-target interactions we constructed a network of our validated miRNAs and their predicted genes targeted by more than two of the miRNAs. Two candidate genes, *CLIP3* and GPR22 were the target of three of the differentially expressed miRNAs (*miR-215*, *miR-34a* and *miR-192*) and 52 candidate genes were targeted by two of the five miRNAs. *CLIP3* encodes for cap-gly domain-containing linker protein 3 (CLIP3) and is associated with the plasma membrane and trans-Golgi membrane [68]. Studies in the rat have shown that CLIP3 is highly expressed in neurones and glial cells and is associated with myelination and nerve tissue regeneration after peripheral nerve injury [69,70]. Accordingly, CLIP3 may have a role in neuronal homeostasis, possibly through the maintenance of polarised protein/membrane intracellular trafficking and cytoskeleton remodelling. *GPR22* encodes for the orphan G-protein-coupled receptor, GPR22, and has been reported to be expressed in many regions of the brain [71]. No ligand has been identified for GPR22 and its biological function remains poorly understood. It is not clear how or if these genes have any involvement in the pathology of MNGIE, however, both appear to be expressed in tissue systems that are affected by the disease and thus warrant further investigation.

## 4. Materials and Methods

### 4.1. Study Design and Subjects

The study was conducted according to the study design depicted in Figure 1. A total of 65 patients with MNGIE and 65 age and sex matched healthy controls were recruited into this study and were assigned to the four study phases: discovery phase, 10 patients and 10 healthy controls; candidate screening phase, 20 patients and 20 healthy controls; validation phase, 35 patients and 35 healthy controls; performance phase, 4 patients who were included in the validation study but subsequently received treatment with EE-TP. Different participants were recruited into the discovery, candidate and validation phases. The inclusion criteria for patient eligibility were a definitive diagnosis of MNGIE due to thymidine phosphorylase deficiency based upon DNA sequencing, and/or < 10% of normal thymidine phosphorylase activity in the buffy coat; and biochemical criteria. Exclusion criteria for both patients and healthy controls included participation in a controlled trial of an investigational medicinal product, receipt of blood transfusions within the past 4 months and a current or past history of hepatitis B, hepatitis C or human immunodeficiency virus infection. Due to the rarity of MNGIE, patient recruitment was performed at a number of international centres. The study was conducted according to the guidelines of the Declaration of Helsinki, and approved by the NHS Research Ethics Committee (London—Surrey Borders, reference 18/LO/2173). Written informed consent was obtained from all participants.

### 4.2. Blood Collection

Twenty mL of venous blood were collected from each participant into one serum (SST) and/or one K_2_EDTA-treated BD Vacutainer tubes (Beckman Dickinson, Berkshire, UK) using a standard phlebotomy protocol. For patients treated with EE-TP, blood was collected prior the administration the treatment cycles listed in Table 6 so as to concur with the clinical observations recorded. Blood collected into serum tubes were left at room temperature for 30 min to permit coagulation, whereas blood collected into anticoagulant was mixed by gently inverting the tubes 10 times and then processed immediately. Both tubes were centrifuged at 1500× *g* for 10 min at 4 °C. Following centrifugation, the serum and plasma supernatants were aspirated using RNase-free pipette tips and apportioned into RNase-free cryotubes as 0.5 mL aliquots and then stored at −80 °C until required for miRNA extraction. Samples were visually inspected for pink colouration and those which indicated haemolysis were excluded from the study. miRNA profiling for the different study phases using qPCR or NGS was then conducted according to the workflow as presented in Figure 14 and described in detail below.

### 4.3. miRNA RT-qPCR

miRNA profiling using RT-qPCR was performed in serum and plasma samples collected for the discovery, validation and performance study phases (Figure 14).

#### 4.3.1. RNA Isolation and Sample Quality Control

Prior to profiling using miRNA qPCR panels, samples were first subjected to a number of quality control checks to ensure the absence of unwanted bias arising from pre-analytical or analytical variables, and to establish that the quality of the RNA input was sufficiently high for effective amplification.

Serum and plasma samples were thawed on ice and then centrifuged at 3000× *g* for 5 min in a 4 °C microcentrifuge to remove cryoprecipitates. Two hundred microlitres of each plasma or serum sample were transferred to a new tube, to which 60 μL of lysis solution BF containing 1 μg carrier-RNA/60μL lysis solution BF and 1 µL RNA spike-in template (UniSp2 and UniSp4 which detect differences in RNA extraction efficiency) were added. After mixing for 1 min and incubation at room temperature for 7 min, 20 μL protein precipitation solution BF was added, and the mixture was vortexed and incubated at room temperature for 1 min. Following centrifugation at 11,000× *g* for 3 min, total RNA was extracted from the supernatants using the miRCURY RNA isolation Kit for Biofluids. The purified RNA was eluted into 50 µL nuclease-free water and stored at −80 °C until analysis.

First-strand cDNA was synthesised from each RNA sample using the miRCURY LNA RT kit (Qiagen, Hilden, Germany) and tested for the expression of the endogenous miRNAs, *miR-103*, *miR-23a*, *miR-30c*, *miR-451* and *miR-142-3p* which are typically expressed in biofluids. The expression of the three-synthetic spike-ins was also examined (UniSp2, UniSp4 and UniSp6). Briefly the RNAs were tailed with a poly(A) sequence at their 3′end and then reverse transcribed into cDNA using a universal poly(T) primer with a 3′end degenerate anchor and a 5′end universal tag; one reaction contained 4 µL of template RNA (5 ng/µL), 4 µL of 5× SYBR Green Reaction Buffer, 2 µL of 10 × RT enzyme, 1 µL RNA spike in (UniSp6 which evaluates the efficiency of the cDNA synthesis reaction for signs of inhibition) and 9 µL nuclease-free water. To detect RNA contamination in the reverse transcription reaction (RT), negative controls excluding template were included in the RT step. RT reactions were incubated in a thermocycler for 60 min at 42 °C and then 5 min at 95 °C to inactivate the reaction. Immediately after incubation, cDNA was diluted 50x by the addition of nuclease-free water. Quantitative PCR (qPCR) was performed in a total volume of 10 μL according to the miRCURY LNA miRNA PCR protocol. Briefly one reaction contained 5 µL 2x SYBR Green Master Mix, 1 µL forward and reverse PCR primer mix, 3 µL of diluted cDNA template and 1 µL nuclease-free water. Amplification was performed in a LightCyler 480 Real-Time PCR system (Roche, Basel, Switzerland) in 384 well plates at 95 °C for 2 min, 45 cycles of 95 °C for 10 s and 56 °C for 60 s, and this was followed by melting curve analysis at 60–95 °C.

The expression levels of the endogenous miRNAs and spike-ins were examined; any sample outliers would be excluded from any further analysis. Sample haemolysis was assessed from the differential of endogenous *miR-23a-3p* and *miR-451*, where *miR-451* is normally expressed within erythrocytes and *miR-23a* is stably expressed in plasma and serum and is not affected by haemolysis. A delta Cq (*miR23a*-*miR451*) value greater than 7 is an indicator haemolysis, and affected samples were removed from downstream analysis. Signals from the no template negative control were assessed and only miRNAs that elicited signals at least five Cq-values lower than the negative control were included in the panel profiling described below. For negative control assays that did not yield any signal, the upper limit of detection was set to Cq = 37.

#### 4.3.2. miRNA Profiling Using RT-qPCR Panels

Samples that passed the quality control checks above were profiled using miRNA qPCR panels as follows. RNA was reverse transcribed in 20 μL reactions using the miRCURY LNA RT Kit (Qiagen, Hilden, Germany) as described above. cDNA was diluted 50 x and assayed in 10 μL PCR reactions according to the protocol for miRCURY LNA miRNA PCR. For the discovery phase, each miRNA was assayed once by qPCR on the miRNA Ready-to-Use PCR, Human panel I+II using miRCURY LNA SYBR Green master mix (752 assays). For the validation and confirmatory phases, each miRNA was assayed once by qPCR on the custom designed panel using miRCURY LNA SYBR Green master mix (95 assays, see Table A4). Negative controls excluding template (no template control) from the reverse transcription reaction were performed and profiled as the samples. Amplification was performed in 384 well plates using the cycling conditions described above. Melting curve analyses were performed at the end of the PCR cycles.

#### 4.3.3. qPCR Data Collection and Quality Control

Data were analysed following the guidelines for Minimum Information for Publication of Quantitative Real-Time PCR Experiments (MIQE) [72]. Amplification curves for each target miRNA were analysed using the Roche LightCyler software to obtain a Cq value, calculated using the method of the second derivative. To determine the specificity of the PCR products, a melting curve was generated and analysed for each assay using the LightCyler software. The appearance of a single peak with the expected melting temperature was an indication that a single specific product was amplified during the qPCR process; any reactions that gave rise to multiple melting curve peaks or single peaks with a melting temperature that was inconsistent with the assay specifications were removed from the data set. The amplification efficiency was calculated using algorithms similar to the Lin Reg software. Only assays with 5Cq less than the negative control were included in the analysis. For assays that did not yield any signal over the negative control, the upper limit of detection was set to Cq = 37.

#### 4.3.4. Normalisation

Raw Cq data that passed the above quality control criteria were then subjected to normalisation to control for differences in the amount of RNA added to each reaction. For the discovery phase, normalisation was performed by first identifying which miRNAs were detected in all samples, and then taking the average of the identified miRNA Cq values for each sample. For this study phase this included 137 miRNA assays, as the stability of the average of these was higher than any single miRNA in the data set as measured by the Normfinder software [73]. For the validation and confirmatory phases, a custom normalisation was applied, based on candidates identified in the miRNA sequencing study (candidate screening phase) by determining the most stable endogenously expressed genes for optimal normalisation. Exponentially transformed Cq values (2−Cq) were imported into the NormFinder software and the stability value of each reference gene was computed, with genes of the lowest stability values indicating a higher consistency of miRNAs across different samples and groups. The average stability values of gene combinations were investigated to assess if the average of these could improve stability with respect to the use of a single gene. The combination of miR-*30e-5p* and *miR-15b-5p* were found to provide the lowest stability value and therefore the average Cq values of these miRNAs for each sample was employed as the normaliser. The following formula was used to calculate the normalised Cq values for each miRNA:

Normalised Cq = average sample Cq (of miRNAs detected in all assays)—assay Cq (specific miRNA in assay). Thus, a higher value would indicate a greater abundance of a specific miRNA in a particular sample.

#### 4.3.5. Differential Gene Expression Analysis

Expression differences (ΔΔCq) were calculated between the ΔCq values of the patient and control samples (ΔCq patient-ΔCqcontrol). The fold change was calculated as 2-ΔΔCq, whereby values >1 were considered as upregulation and those values <1 were considered as downregulation. Significant miRNAs between the control and patient group were calculated using the two-tailed Student’s *t* test, with *p*-values then adjusted for false discovery rate (FDR) via the Benjamini–Hochberg method. Statistical significance was then gauged by any variable that passed the adjusted *p*-value ≤ 0.05.

### 4.4. miRNA NGS Analysis

miRNA sequencing analysis was performed on serum samples collected for the candidate screening phase, according to the workflow presented in Figure 14.

#### 4.4.1. RNA Isolation and Quality Control

Prior to library preparation, samples were first subjected to quality control steps using qPCR to check for sample haemolysis, cDNA synthesis inhibition, RNA isolation efficiency and to confirm that sample miRNA expression levels were within the expected range.

Plasma samples were thawed on ice, centrifuged at 3000× *g* for 5 min at 4 °C and then processed for RNA isolation using the miRNeasy Serum/Plasma Kit (Qiagen, Hilden, Germany). Briefly, 1 mL QIAzol Lysis Reagent and 200 µL of plasma sample were mixed and incubated at room temperature for 5 min before adding 3.5 µL miRNeasy Serum/Plasma spike-in control containing UniSp 100 and UniSp 101 for the assessment of RNA isolation efficiency, and UniSP132, UniSP133, UniSP135, UniSP137, UniSP138, UniSP140 and UniSP142 for assessing the reproducibility and linearity of the NGS reads downstream. Two hundred µL of chloroform were added to each lysate and the tubes thoroughly mixed by vortexing for 15 s, followed by incubation at room temperature for 3 min and then centrifugation at 12,000× *g* at 4 °C for 15 min. The upper phase containing the RNA was transferred to a new collection tube to which 1.5 volumes of 100% ethanol were added and the sample mixed thoroughly by pipetting. RNA was purified from 700 µL of each sample using RNeasy MinElute spin column and eluted into a final volume of 12 µL nuclease-free water.

First-strand (cDNA) was synthesised from each RNA sample using the miRCURY LNA RT kit (Qiagen, Hilden, Germany) and tested for the expression of *miR-103*, *miR-23a*, *miR-30c*, *miR*-*451*, *miR-142-3p* and the three-synthetic spike-ins (UniSp 100, UniSp 101 and UniSp 6) as described above, but using half the volumes of the reaction components. qPCR was performed in a total volume of 100 µL using 50 µL 2x miRCURY SYBR Green Master Mix, 1 µL of undiluted cDNA template and 49 µL RNase-free water. Amplification was performed in 384 well plates using the cycling conditions as described above.

#### 4.4.2. Library Preparation

The library was prepared using the QIAseq miRNA Library Kit (Qiagen, Hilden, Germany), where 5 µL of total RNA obtained using the miRNeasy Serum/Plasma Kit were converted to miRNA NGS libraries. Briefly, a pre-adenylated DNA adapter was ligated to the 3′ ends of miRNAs, followed by ligation of an RNA adapter to the 5′ end. A reverse-transcription primer containing an integrated UMI was used to convert the total 3′/5′ ligated miRNAs into cDNA. Following cDNA purification using magnetic beads, the library was amplified with indexing forward primers and a universal reverse primer using 22 cycles of PCR and then cleaned-up using magnetic beads.

#### 4.4.3. Library Quality Control

Pre-sequencing library quality control and concentrations were performed by analysing 1 µL of the miRNA sequencing library on an Bioanalyzer 2100 (Agilent, California, USA) using a using a high sensitivity DNA chip. The concentration of the library was then determined in 2 µL using a Qubit Fluorimeter (Thermo Fisher Scientific, Massachusetts, USA) with the dsDNA HS Assay Kit (Table A1).

#### 4.4.4. NGS

Libraries that passed the quality control check were pooled in equimolar ratios and sequenced on a NextSeq500 instrument as a 75 bp read length (up to 46 bp insert + 19 bp 3′ linker + 10 bp UMIs) with an average depth of 22 million reads per sample.

#### 4.4.5. Raw Data Processing

The workflow for miRNA sequencing data analysis, including trimming, alignment, quantitation, normalisation and differential gene expression analysis is shown in Figure 15.

Raw data were de-multiplexed and FASTQ files were generated for each sample using the bcl2fastq software (Illumina Inc., San Diego, CA, USA). FASTQ data were quality control checked using the FastQC tool and if identified, low-quality reads and artefacts were removed. To correct PCR bias with UMI information, Cutadapt software (version 1.11) was used to extract information on the adaptor presence and UMI raw reads. The output was used to remove adapter sequences and to collapse reads by UMI. Briefly, reads were processed by trimming off the 3′ adapter and low-quality bases. This was followed by identification of insert sequences and UMI sequences. Reads with less than 16 bp insert sequences (too short) or less than 10 bp UMI sequences (UMI defective) were discarded. All reads containing identical insert sequence and UMI sequence combinations (insert-UMI pair) were collapsed into a single read. The output of UMI correction included insert sequences from collapsed full-UMI reads and reads which did not contain full length UMI sequence (partial-UMI reads). These were then subjected to quality control checks using FastQC to examine the overall sequence quality, the guanine-cytosine percentage distribution and the presence of overrepresented sequences.

Reads that passed the above checks were then mapped using Bowtie 2 (2.2.2) as follows: reads were first aligned to spike-ins (added at the RNA extraction stage, to monitor extraction efficiency) and abundant sequences (outmapped reads, including polyA and polyC homopolymers and abundant ribosomal or mitochondrial RNA sequences) were filtered out. Perfect match to the reference sequence was required. The remaining reads were then aligned to mature sequences of miRBase 20, with the requirement of a perfect match. Unmapped reads were then aligned to the human reference genome, GRCh37, with not more than one mismatch permitted in the first 32 bases of the read. Reads which aligned to known miRNA loci in the genome were combined with the miRBase-mapped reads. Reads that mapped to the reference genome outside the miRNA locus were classified as novel miRNAs and were predicted using mirPara64 and miRbase. No indels were allowed in any of the mapping steps.

To determine the miRNA call rate, after mapping the reads for a particular miRNA were divided by the total number of mapped reads in the sample and then multiplied by one million, denoted as TPM. For the purpose of differential expression analysis, raw reads of the samples were normalised using Trimmed Mean of M values (TMM) normalisation method using the EdgeR statistical software package (Bioconductor, http://bioconductor.org/, accessed on 20 March 2021).

#### 4.4.6. Differential Gene Expression Analysis

Differential expression analysis was performed using the EdgeR statistical software package (Bioconductor, http://bioconductor.org/, accessed on 20 March 2021). To determine dysregulated miRNAs associated with MNGIE, the TMM normalised miRNA expression profiles of the disease samples were compared to the healthy control samples (disease vs. healthy). A log_2_ Fold Change (logFC) in gene expression level with a change cut-off of 1, representing the size of the change and a *p*-value with a false discovery rate (FDR) correction cut off of < = 0.05 (Benjamini and Hochberg), representing the significance of the change, were used to obtain the differentially expressed miRNAs [74]. Separate comparisons were also made between the following three disease and healthy age groups to identify miRNAs that maybe associated with disease progression: ≤19, 20–29 and ≥30 years of age.

### 4.5. Downstream Bioinformatics Analyses

All downstream analyses were conducted in R Programming Language (R Core Team, 2016).

#### 4.5.1. General Bioinformatics and QC

Unsupervised sample clustering was performed through the generation of dendrograms using Euclidean distance and Ward’s linkage, using the dist and hclust functions, respectively. The distribution of samples was checked via box and whisker and violin plots using ggplot2 [75]. QC was also gauged via pairwise scatterplot comparisons between control and MNGIE samples using the scatterplotMatrix function from the car package. Linear regression, non-parametric gamma, and LOESS regression lines were fitted across distributions and a histogram generated for each sample along the diagonal of the plot. PCA was performed via the PCAtools package and volcano plots were generated via the EnhancedVolcano package (https://github.com/kevinblighe/EnhancedVolcano, accessed on 20 March 2021). Supervised clustering was performed by filtering genes/miRNAs based on our threshold for statistical significance. Input data were converted to the Z scale and then clustered via 1—Pearson correlation distance and Ward’s linkage using the Heatmap function of the ComplexHeatmap package [76]. Box and whisker plots of miRNA Z-scores were added to the heatmap diagram. Network plots and graphs were generated via the igraph package [69]. Finally, box and whisker plots with scatter plot overlays were produced via ggplot2.

#### 4.5.2. Candidate Phase Micro RNA Signature

In the candidate phase, an initial miRNA signature (80 miRNAs) was identified by taking those miRNAs that reached statistical significance from the following comparisons:

Disease group 1 vs. healthy group 1;

Disease group 2 vs. healthy group 2;

Disease group 3 vs. healthy group 3;

Disease vs. healthy.

To refine this signature, the identified miRNAs were used in a multinomial elastic-net penalised regression model via glmnet [77]; this was cross-validated 10-fold and key miRNAs identified by taking those whose coefficients were not shrunk to zero. The resulting refined panel (*n* = 41) was used in ROC analyses to derive the AUC. This initial signature of 80 miRNAs (plus 10 further miRNAs of interest) was used to design a qPCR panel for the validation phase.

#### 4.5.3. Validation Phase

In the validation phase, fold-change comparisons were illustrated via ggplot2 [75], comparing discovery vs. validation log_2_ fold-change values. A meta-analysed combined *p*-value was calculated for each miRNA via the CombP function from GenRank in R; this was performed separately for serum miRNA-seq/plasma qPCR and serum miRNA-seq/serum qPCR. The meta-analysed *p*-value was then used to define a final panel of miRNAs for the overall study. These final miRNAs were then further tested by regressing each independently to the main end point (disease versus healthy) in a binary logistic regression model. AUC was gauged by ROC analysis by plotting sensitivity against specificity using the pROC package [78]. An area greater than 0.5 under the curve suggests the utility of the miRNA in question in discriminating between healthy and disease.

#### 4.5.4. MirRNA Gene Target Identification

In order to identify gene targets of each statistically significant miRNA, miRNAtap was employed [79]. The following databases were selected for searching predicted gene targets: DIANA, PicTar, TargetScan and miRanda [80,81,82,83]. The geometric mean score was used for ranking each gene. A gene target for a particular miRNA was required to be predicted in at least 2 of the 4 databases in order to be included. The final list of gene targets was then visualised via the oncoPrint function of the ComplexHeatmap package [76].

#### 4.5.5. Enrichment and Pathway Analysis

Target genes identified from miRNAtap were fed into topGO and KEGGprofile for gene enrichment and pathway analysis, respectively [84,85]. The classic algorithm with the Kolmogorov-Smirnov (K-S) test, which takes the rank of each target gene into consideration, was used to determine statistically significantly enriched terms for each miRNA’s target genes. For KEGG pathway enrichment, FDR adjusted *p*-values were used to determine statistical significance.

## 5. Conclusions

This study shows that five plasma and three serum miRNAs can discriminate MNGIE disease from healthy controls and also the stage of disease. The signal in plasma appears to be superior to that of serum, with the single best predictor of MNGIE being *miR-34a-5p,* having an AUC of 0.977, and it was shown to be elevated across each age sub-group of disease, when compared to any healthy group. A decrease in expression of *miR-34a-5p* was noted during treatment with EE-TP in all four patients and coincided with clinical biochemical and/or clinical improvements. Enrichment and pathway analysis identified a number of candidate genes and networks that may play roles in MNGIE and are worthy of further investigation. Although the dysregulation of these miRNAs is unlikely to be specific to MNGIE compared to other mitochondrial diseases, the availability of an effective and relatively non-invasive biomarker would meet an unmet need. The inclusion of the exploratory plasma miRNA biomarker panel in future clinical trials of investigational therapies for MNGIE would enable an appraisal of its prognostic value in assessing clinical status.

## Figures and Tables

**Figure 1 ijms-22-03681-f001:**
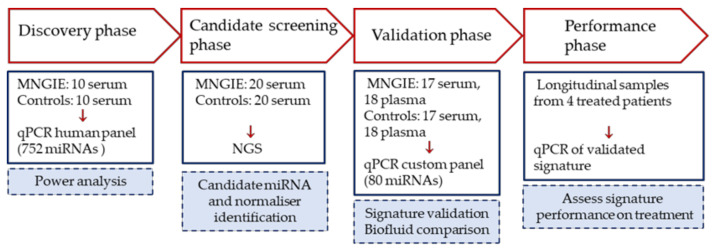
Study design showing the aims of the four study phases, the patient cohort numbers, the sample types analysed and the microRNA (miRNA) analysis platform employed for each phase.

**Figure 2 ijms-22-03681-f002:**
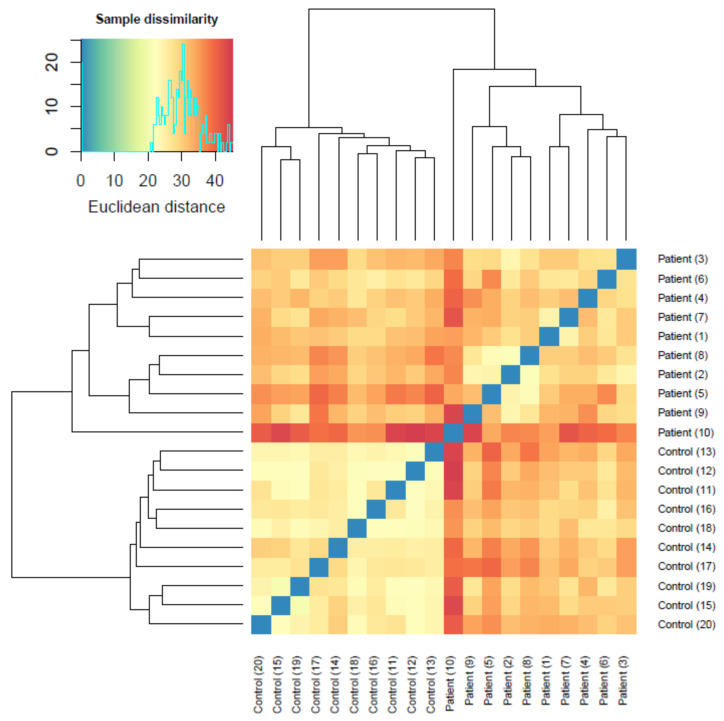
Heatmap of Euclidian distances between healthy control and mitochondrial neurogastrointestinal encephalomyopathy (MNGIE) disease serum miRNA expression profiles (*n*=20). The colour level is proportional to the value of the dissimilarity, ranging from blue (similarity) to red, which corresponds to the highest value of Euclidean distance and thus dissimilarity. The miRNA profiles were more dissimilar between the MNGIE samples.

**Figure 3 ijms-22-03681-f003:**
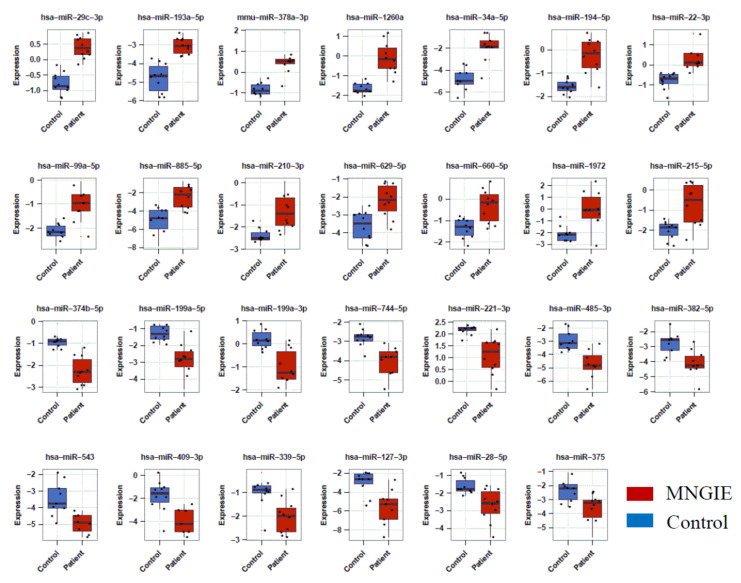
Box plots with scatter-plot overlays of serum miRNA expression in patients with MNGIE and healthy controls. Fourteen miRNAs (top two panels) were significantly upregulated and thirteen miRNAs were significantly downregulated (bottom two panels) in patients with MNGIE compared to healthy controls. Boxes display interquartile ranges (25–75%) and error bars indicate minimum and maximum values.

**Figure 4 ijms-22-03681-f004:**
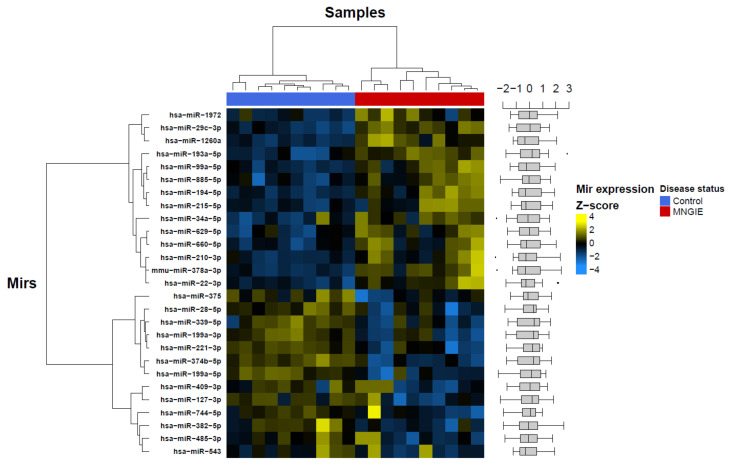
Heatmap of differentially expressed miRNAs determined in the discovery study by quantitative PCR (qPCR) in Scheme 27. statistically significantly differentially expressed miRNAs passing the threshold of *p* ≤ 0.05 (Benjamini–Hochberg adjusted) and an absolute fold-change > ±2. Based on the heatmap shading, the majority of the miRNAs were either exclusively highly expressed in the healthy control serum or in the MNGIE disease serum. A box and whisker plot of miRNA Z-scores is shown on the right-hand side of the heatmap.

**Figure 5 ijms-22-03681-f005:**
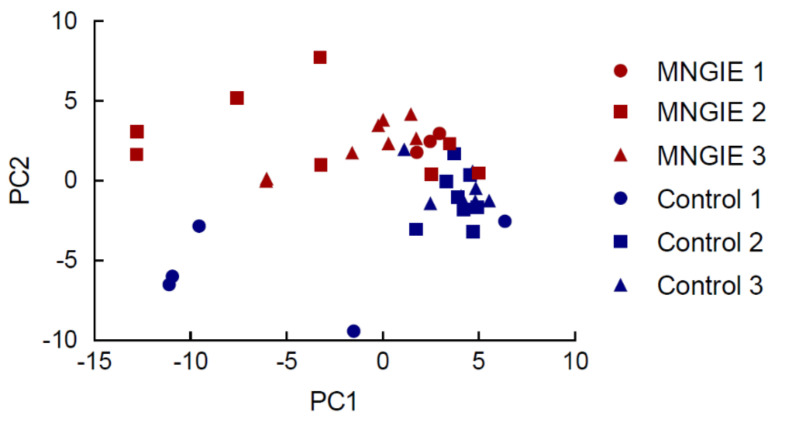
Assessment of intergroup and intragroup variability: A principal component analysis plot displaying all 39 samples in principal component 1 (PC1) and PC2, which describe 25.8% and 8.5% of the variability, respectively, within the expression data set. PC analysis was applied to normalised (TMM) and log-transformed count data. A clear division can be seen in the miRNA expression profiles between the MNGIE and healthy control groups along PC2. Age group 1 = ≤19 years (*n* = 4 and 5 for patient and healthy control groups, respectively); age group 2 = 20–29 years (*n* = 8 for each group); age group 3 = ≥30 years (*n* = 7 for each participant group).

**Figure 6 ijms-22-03681-f006:**
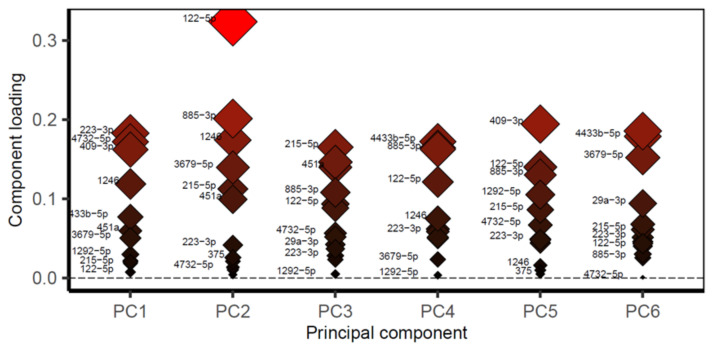
Component loadings for PCs 1–6 reveal that *miR-122-5p* is the dominant source of variation along PC2, a PC along which healthy and disease samples are visually segregated. For each PC, only miRNAs falling within the top 1% of the overall loadings range for each PC were plotted.

**Figure 7 ijms-22-03681-f007:**
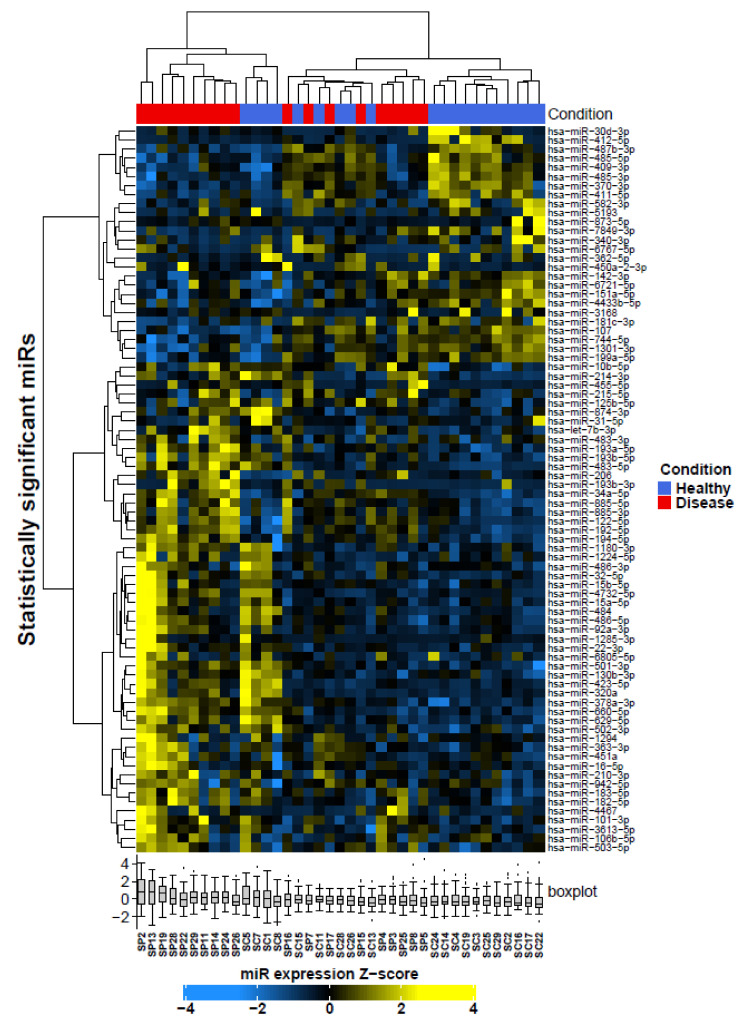
Heatmap and hierarchical cluster analysis of the differentially expressed miRNAs between MNGIE patients and healthy controls detected in the candidate screening study by next generation sequencing (NGS). Supervised clustering was performed on the 80 statistically significantly differentially expressed miRNAs passing the threshold of *p* ≤ 0.05 (Benjamini–Hochberg adjusted) and an absolute fold-change ≥ 1. Rows represent miRNAs and columns represent samples and groups. A box and whisker plot of miRNA Z-scores is shown at the bottom of the heatmap.

**Figure 8 ijms-22-03681-f008:**
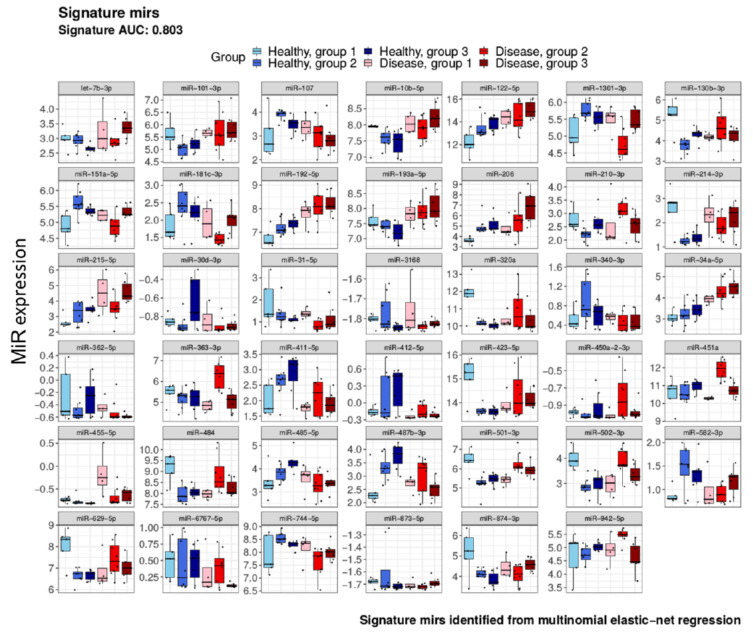
Refined miRNA signature consisting of 41 miRNAs identified after conducting a multinomial elastic-net regression analysis of the 80 differentially expressed miRNAs. These miRNAs were selected based on their model coefficients not being shrunk to zero after cross-validation of the model fit.

**Figure 9 ijms-22-03681-f009:**
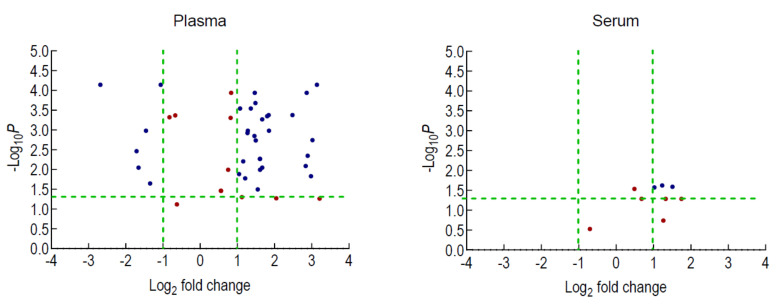
Volcano plots of differentially expressed miRNAs in plasma (left panel) and serum (right panel) showing the relationship between expression fold change and statistical significance. The vertical lines correspond to a 2-fold change in up and down expression, while the horizontal line represents a *p*-value of 0.05. The red points represent miRNAs with no statistical significance after Benjamini–Hochberg correction and a fold change < 2 or > -2, whereas the blue points represent statistically significant up or downregulated miRNAs.

**Figure 10 ijms-22-03681-f010:**
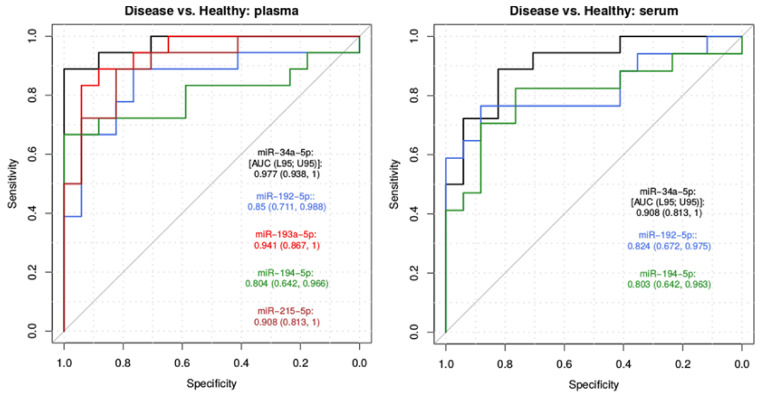
Receiver operating characteristic (ROC) curve analyses to evaluate the diagnostic power of plasma and serum miRNAs. Plasma *miR-34a-5p*, *miR-193a-5p*, *miR-215-5p*, *miR-192-5p* and *miR-194-5p*, healthy controls vs. MNGIE patients (left panel); and serum *miR-34a-5p*, *miR-192a-5p* and *miR-194-5p*, healthy controls vs. MNGIE patients (right panel). An area under curve (AUC) of 0.5 is considered significant.

**Figure 11 ijms-22-03681-f011:**
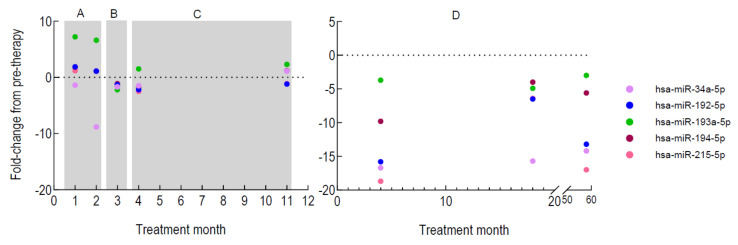
miRNA panel expression fold-changes during treatment with erythrocyte encapsulated thymidine phosphorylase (EE-TP) compared to baseline expression levels for patients A, B and C (left panel) and patient D (right panel).

**Figure 12 ijms-22-03681-f012:**
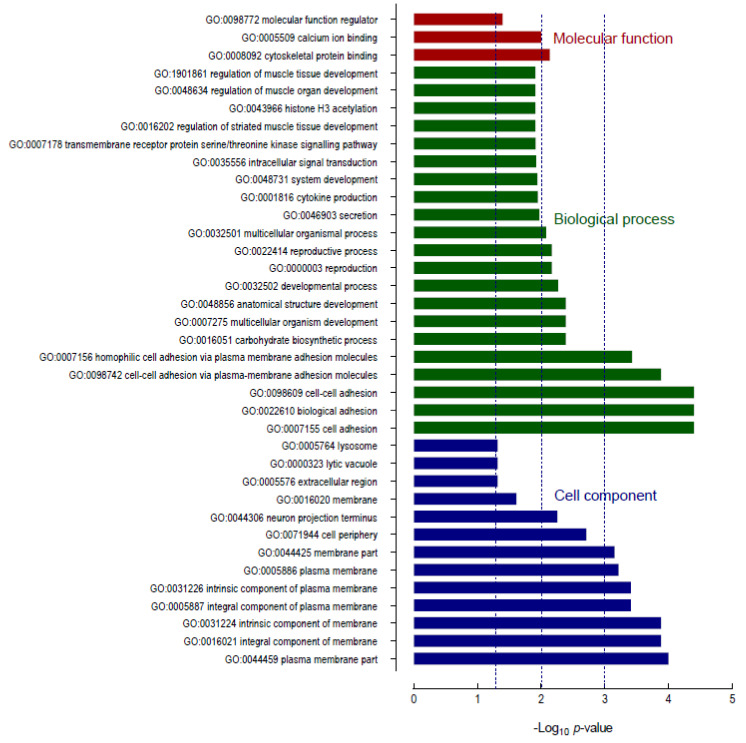
Gene Ontology (GO) analysis of the target genes for the five differentially expressed miRNAs. Top panel: significantly associated terms for molecular functions; middle panel: top 20 significantly associated terms for biological processes; and bottom panel: significantly associated terms for cell components. The terms are ordered by Kologorov–Smirnov *p*-value (−log) with the *x*-axis showing the –log_10_
*p* value for each GO term. Vertical lines from left to right show equivalents of *p* = 0.05, *p* = 0.01 and *p* = 0.0001.

**Figure 13 ijms-22-03681-f013:**
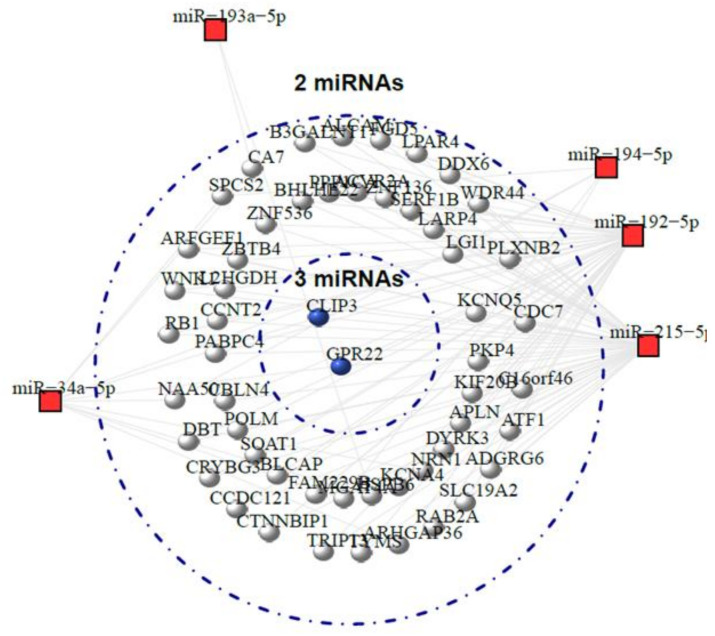
MicroRNA-target gene network for the upregulated miRNA panel in MNGIE. The network shows the relations of the five upregulated miRNAs and their potential target genes. The outer ring represents genes that are targeted by 2 miRNAs. The inner ring containing genes that are targeted by 3 of the miRNAs.

**Figure 14 ijms-22-03681-f014:**
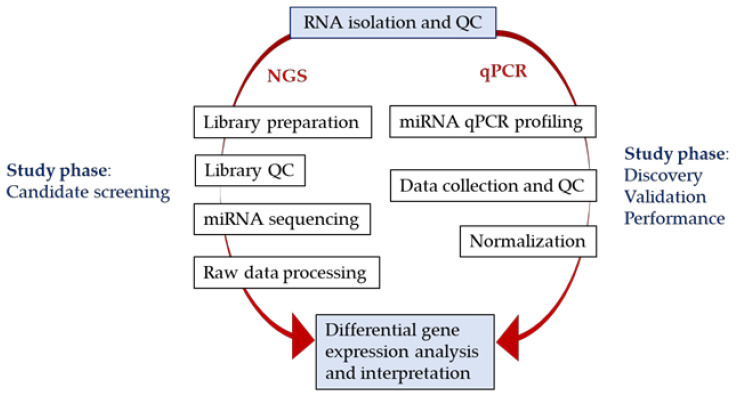
Schematic diagram of miRNA profiling workflow for the different study phases using NGS and qPCR.

**Figure 15 ijms-22-03681-f015:**
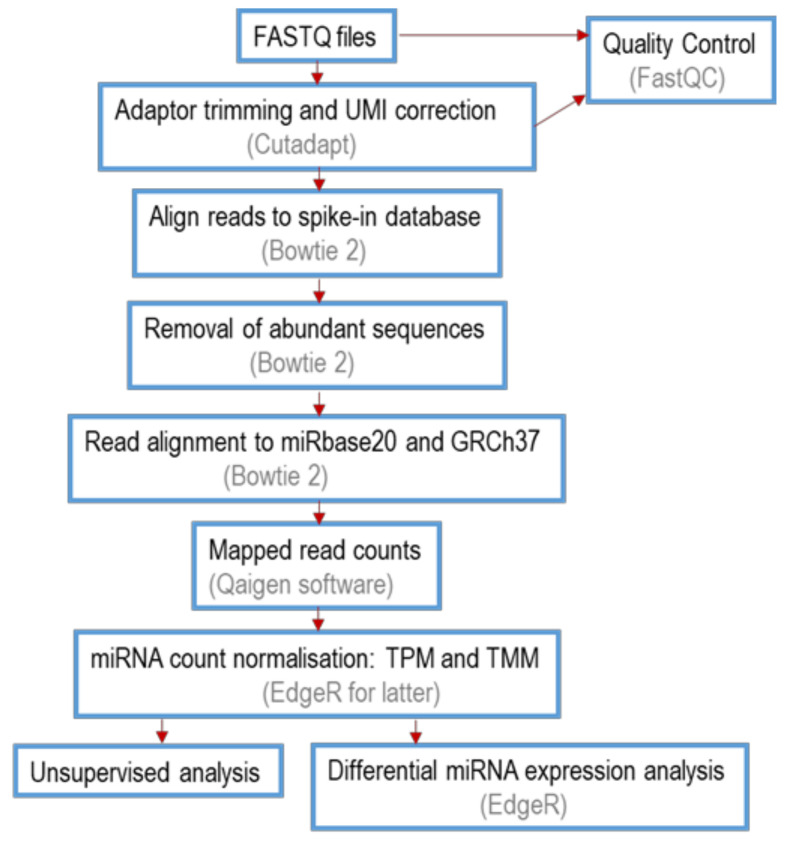
Key steps in the computational workflow for the analysis of miRNA sequencing data.

**Table 1 ijms-22-03681-t001:** Discovery study: fold changes and *p*-values of miRNAs differentially expressed in the sera from healthy controls and patients with MNGIE.

Differentially Expressed miRNA	*p* Value	Fold Change
hsa-miR-29c-3p	0.0002	2.2
hsa-miR-193a-5p	0.0029	3.3
hsa-miR-378a-3p	0.0029	2.6
hsa-miR-1260a	0.0039	3.0
hsa-miR-34a-5p	0.0043	7.1
hsa-miR-194-5p	0.0064	2.5
hsa-miR-22-3p	0.0074	2.3
hsa-miR-99a-5p	0.0097	2.2
hsa-miR-885-5p	0.0110	5.8
hsa-miR-210-3p	0.0215	2.1
hsa-miR-629-5p	0.0226	2.6
hsa-miR-660-5p	0.0226	2.1
hsa-miR-1972	0.0226	4.1
hsa-miR-215-5p	0.0278	2.4
hsa-miR-374b-5p	0.0029	-2.6
hsa-miR-199a-5p	0.0104	-3.0
hsa-miR-199a-3p	0.0110	-2.3
hsa-miR-744-5p	0.0137	-2.3
hsa-miR-221-3p	0.0211	-2.1
hsa-miR-485-3p	0.0219	-3.5
hsa-miR-382-5p	0.0226	-2.5
hsa-miR-543	0.0233	-2.8
hsa-miR-409-3p	0.0255	-4.6
hsa-miR-339-5p	0.0255	-2.1
hsa-miR-127-3p	0.0278	-6.1
hsa-miR-28-5p	0.0376	-2.2
hsa-miR-375	0.0472	-2.4

**Table 2 ijms-22-03681-t002:** Differentially expressed miRNAs detected by sequencing in sera from patients with MNGIE compared to sera of healthy controls, according to age range.

Age Range *	miRNA ID	MNGIE Mean TMM	Healthy Control Mean TMM	Log FC	*p*-Value	FDR
*miRNAs over-expressed in MNGIE* vs. *healthy controls*						
≤19	hsa-miR-215-5p	183	6	4.60	<0.0001	0.0211
20–29	hsa-miR-451a	14,019	4301	1.70	<0.0001	0.0004
	hsa-miR-32-5p	227	39	2.52	<0.0001	0.0028
	hsa-miR-363-3p	315	99	1.66	<0.0001	0.0028
	hsa-miR-501-3p	274	91	1.56	<0.0001	0.0028
	hsa-miR-194-5p	588	203	1.53	<0.0001	0.0028
	hsa-miR-502-3p	69	18	1.89	0.0001	0.0028
	hsa-miR-4732-5p	1927	320	2.59	0.0001	0.0029
	hsa-miR-214-3p	14	1	2.84	0.0001	0.0030
	hsa-miR-210-3p	34	8	1.98	0.0001	0.0030
	hsa-miR-107	866	236	1.88	0.0001	0.0030
	hsa-miR-22-3p	1239	362	1.77	0.0001	0.0040
	hsa-miR-34a-5p	82	21	1.93	0.0002	0.0060
	hsa-miR-106b-5p	43	13	1.68	0.0002	0.0060
	hsa-miR-942-5p	149	69	1.09	0.0002	0.0060
	hsa-miR-192-5p	991	376	1.40	0.0003	0.0075
	hsa-miR-486-5p	262,731	103,131	1.35	0.0004	0.0086
	hsa-miR-130b-3p	108	33	1.69	0.0004	0.0089
	hsa-miR-16-5p	360,834	151,980	1.25	0.0005	0.0099
	hsa-miR-15a-5p	800	256	1.64	0.0006	0.0108
	hsa-miR-660-5p	751	332	1.17	0.0006	0.0110
	hsa-miR-1285-3p	7	0	4.77	0.0008	0.0121
	hsa-miR-484	1831	647	1.50	0.0007	0.0121
	hsa-miR-182-5p	1745	632	1.46	0.0008	0.0121
	hsa-miR-378a-3p	247	121	1.02	0.0010	0.0146
	hsa-miR-183-5p	1230	459	1.42	0.0017	0.0225
	hsa-miR-1224-5p	83	19	2.16	0.0019	0.0237
	hsa-miR-320a	9249	3080	1.59	0.0019	0.0237
	hsa-miR-629-5p	618	261	1.24	0.0019	0.0237
	hsa-miR-503-5p	55	23	1.23	0.0023	0.0258
	hsa-miR-92a-3p	53,636	25,118	1.09	0.0023	0.0258
	hsa-miR-3613-5p	182	57	1.66	0.0028	0.0296
	hsa-miR-483-5p	1689	579	1.54	0.0030	0.0296
	hsa-miR-4467	13	1	3.15	0.0033	0.0323
	hsa-miR-423-5p	82,538	33,034	1.32	0.0035	0.0330
	hsa-miR-6805-5p	16	3	2.21	0.0047	0.0404
	hsa-miR-450a-2-3p	4	0	2.50	0.0049	0.0408
	hsa-miR-101-3p	1911	886	1.11	0.0051	0.0420
	hsa-miR-1294	70	30	1.20	0.0053	0.0422
	hsa-miR-15b-5p	1428	703	1.02	0.0053	0.0422
	hsa-miR-1180-3p	232	98	1.24	0.0060	0.0452
	hsa-miR-486-3p	225	102	1.13	0.0065	0.0460
	hsa-miR-885-5p	60	13	2.15	0.0071	0.0494
≥30	hsa-miR-483-5p	2391	478	2.32	<0.0001	0.0031
	hsa-miR-215-5p	142	37	1.97	0.0001	0.0034
	hsa-miR-34a-5p	108	31	1.75	0.0001	0.0034
	hsa-let-7b-3p	45	14	1.60	0.0001	0.0034
	hsa-miR-192-5p	1150	487	1.24	0.0001	0.0043
	hsa-miR-193b-3p	9	0	4.84	0.0001	0.0053
	hsa-miR-193b-5p	199	51	1.96	0.0002	0.0053
	hsa-miR-193a-5p	1001	409	1.29	0.0002	0.0054
	hsa-miR-214-3p	27	5	2.56	0.0003	0.0084
	hsa-miR-206	928	145	2.67	0.0004	0.0085
	hsa-miR-125b-5p	2115	989	1.09	0.0003	0.0085
	hsa-miR-194-5p	618	259	1.25	0.0006	0.0128
	hsa-miR-122-5p	160,505	52,033	1.63	0.0007	0.0137
	hsa-miR-10b-5p	1093	514	1.09	0.0007	0.0137
	hsa-miR-885-3p	156	40	1.94	0.0010	0.0202
	hsa-miR-874-3p	86	38	1.19	0.0020	0.0346
	hsa-miR-4467	15	2	2.86	0.0026	0.0412
	hsa-miR-423-5p	59,588	33,945	0.81	0.0028	0.0432
*miRNAs under-expressed in MNGIE* vs. *healthy controls*						
20–29	hsa-miR-181c-3p	2	26	−3.09	<0.0001	0.0004
	hsa-miR-1301-3p	66	172	−1.36	<0.0001	0.0028
	hsa-miR-142-3p	3476	7917	−1.19	0.0002	0.0050
	hsa-miR-5193	0	7	−3.66	0.0002	0.0060
	hsa-miR-744-5p	546	1169	−1.10	0.0005	0.0106
	hsa-miR-582-3p	3	15	−2.32	0.0006	0.0108
	hsa-miR-3168	0	6	−3.37	0.0008	0.0121
	hsa-miR-151a-5p	84	182	−1.09	0.0013	0.0179
	hsa-miR-873-5p	0	4	−3.85	0.0028	0.0296
	hsa-miR-340-3p	3	12	−2.16	0.0037	0.0330
	hsa-miR-7849-3p	1	6	−2.55	0.0044	0.0385
	hsa-miR-4433b-5p	327	949	−1.54	0.0059	0.0450
	hsa-miR-31-5p	2	11	−2.21	0.0061	0.0456
	hsa-miR-6721-5p	10	32	−1.62	0.0065	0.0460
≥30	hsa-miR-487b-3p	11	64	−2.42	<0.0001	0.0005
	hsa-miR-370-3p	55	235	−2.07	<0.0001	0.0013
	hsa-miR-485-3p	114	492	−2.12	<0.0001	0.0013
	hsa-miR-485-5p	28	87	−1.65	0.0001	0.0034
	hsa-miR-412-5p	0	9	−3.57	0.0001	0.0051
	hsa-miR-411-5p	7	41	−2.35	0.0002	0.0053
	hsa-miR-6767-5p	0	6	−4.30	0.0004	0.0085
	hsa-miR-362-5p	0	5	−4.15	0.0011	0.0203
	hsa-miR-409-3p	626	1788	−1.51	0.0017	0.0307
	hsa-miR-30d-3p	0	6	−2.91	0.0031	0.0458
	hsa-miR-199a-5p	31	67	−1.02	0.0035	0.0498

* Sample size: ≤19, *n* = 4 for healthy control group, *n* = 5 for patient group; 20–29, *n* = 8 for both healthy control and patient groups; ≥30; *n* = 7 for healthy control and patient groups.

**Table 3 ijms-22-03681-t003:** The ten most stably expressed miRNAs across all samples analysed by NGS. Stability was calculated by Normfinder. A low stability value indicates good stability. The tags per million mapped reads (TPM) is an estimate of the abundance of the miRNA across the MNGIE and healthy control groups.

Name	Stability/Mean TPM	Stability	Mean TPM
hsa-miR-30e-5p	0.142288503	262.38	1844
hsa-miR-425-5p	0.277956693	706.01	2540
hsa-let-7i-5p	0.296527468	1597.69	5388
hsa-let-7b-5p	0.306335853	11,346.68	37,040
hsa-miR-148a-3p	0.31205228	1122.14	3596
hsa-miR-142-5p	0.334079031	524.17	1569
hsa-miR-21-5p	0.341387813	969.2	2839
hsa-let-7a-5p	0.35729798	8206.42	22,968
hsa-miR-146a-5p	0.361150677	1227.19	3398
hsa-miR-93-5p	0.374529201	942.69	2517

**Table 4 ijms-22-03681-t004:** Differentially expressed miRNAs detected by qPCR in the plasma and sera of patients with MNGIE compared to healthy controls.

Biofluid	miRNA	Log_2_ Fold Change	Regulation	Benjamini–Hochberg Adjusted *p*-Value
Plasma	hsa-miR-34a-5p *	3.14095	Up	0.000072
	hsa-miR-142-3p	−2.68394	Down	0.000072
	hsa-miR-107 *	−1.05870	Down	0.000072
	hsa-miR-363-3p *	1.47048	Up	0.000115
	hsa-miR-193a-5p *	2.86444	Up	0.000115
	hsa-miR-423-5p *	1.48452	Up	0.000209
	hsa-miR-660-5p	1.36373	Up	0.000285
	hsa-miR-92a-3p	1.07172	Up	0.000285
	hsa-miR-378a-3p	1.84138	Up	0.000419
	hsa-miR-4732-5p	2.48124	Up	0.000419
	hsa-miR-501-3p *	1.80138	Up	0.000449
	hsa-miR-486-5p	1.66803	Up	0.000540
	hsa-miR-502-3p *	1.28134	Up	0.001049
	hsa-miR-215-5p *	1.84928	Up	0.001049
	hsa-miR-411-5p *	−1.45695	Down	0.001049
	hsa-miR-320a *	1.27269	Up	0.001196
	hsa-miR-629-5p *	1.46049	Up	0.001413
	hsa-miR-193b-3p	3.02042	Up	0.001799
	hsa-miR-10b-5p *	1.4974	Up	0.001838
	hsa-miR-370-3p	−1.70755	Down	0.003432
	hsa-miR-885-5p	2.89025	Up	0.004502
	hsa-miR-192-5p *	1.60991	Up	0.005376
	hsa-miR-183-5p	1.60648	Up	0.005376
	hsa-miR-874-3p *	1.15817	Up	0.006205
	hsa-miR-122-5p *	2.83981	Up	0.008159
	hsa-miR-194-5p	1.67026	Up	0.008929
	hsa-miR-340-3p *	−1.65444	Down	0.008929
	hsa-miR-1180-3p	1.61073	Up	0.010158
	hsa-let-7b-3p *	1.04539	Up	0.013040
	hsa-miR-483-5p	2.97892	Up	0.014706
	hsa-miR-486-3p	1.21158	Up	0.016641
	hsa-miR-199a-5p	−1.34595	Down	0.022482
	hsa-miR-1285-3p	1.54743	Up	0.031837
Serum	hsa-miR-215-5p	1.23468	Up	0.023866
	hsa-miR-34a-5p	1.50986	Up	0.025611
	hsa-miR-192-5p	1.02344	Up	0.026616

* miRNAs common to the refined miRNA signature established following the construction of a multinomial elastic-net penalised regression model of the sequencing data.

**Table 5 ijms-22-03681-t005:** Combined *p*-values and ranking of the top 10 miRNAs after meta-analysis of MNGIE versus healthy control considering serum miRNA-seq and qPCR (plasma and serum).

Plasma			Serum		
miR	Combined *p*-Value	Rank	miR	Combined *p*-Value	Rank
hsa-miR-34a-5p	1.38598 × 10^−9^	1	hsa-miR-34a-5p	3.75631 × 10^−7^	1
hsa-miR-192-5p	1.68099 × 10^−7^	2	hsa-miR-192-5p	7.64172 × 10^−7^	2
hsa-miR-193a-5p	2.07056 × 10^−6^	3	hsa-miR-194-5p	2.61277 × 10^−5^	3
hsa-miR-194-5p	5.05358 × 10^−6^	4	hsa-miR-215-5p	0.000294246	4
hsa-miR-215-5p	1.64213 × 10^−5^	5	hsa-miR-122-5p	0.00250717	5
hsa-miR-22-3p	0.000128536	6	hsa-miR-193a-5p	0.004645046	6
hsa-miR-10b-5p	0.000169857	7	hsa-miR-193b-3p	0.012087665	7
hsa-miR-363-3p	0.000235272	8	hsa-miR-485-3p	0.019741786	8
hsa-miR-107	0.000294182	9	hsa-miR-885-5p	0.022468769	9
hsa-miR-122-5p	0.000473464	10	hsa-miR-10b-5p	0.035076107	10

**Table 6 ijms-22-03681-t006:** Clinical observations and plasma metabolite concentrations at baseline and during treatment with EE-TP and at the time of miRNA assessment.

Patient	Treatment month (Age at Start of Therapy)	EE-TP Dose (U/kg/4 wks)	Plasma Metabolites (µmol/L)		Clinical Observations
			Thy	dUrd	
A	Pre-therapy (37 years)	-	10.5	18.8	Sensorimotor polyneuropathy, external ophthalmoplegia, leukoencephalopathy, intestinal dysmotility with cachexia. Weight:40.6 kg
	1	129	0.5	0.3	No changes observed
	2	129	1.1	4.1	Greater appetite, experienced tingling sensations in feet compared to no sensation pre-therapy, improved swallowing and dysgeusia, tongue less glossitic. Weight: 40.6 kg
B	Pre-therapy (26 years)	-	12	19	Sensorimotor polyneuropathy, external ophthalmoplegia, intestinal dysmotility, anorexia and cachexia. Weight: 31.7 kg
	3	108	2.1	3.9	No changes observed
C	Pre-therapy (25 years)	-	9.0	19.0	Sensorimotor polyneuropathy, external ophthalmoplegia, intestinal dysmotility, anorexia and cachexia. Weight:32 kg
	4	9	5.09	16.0	Reduction in nausea and vomiting Weight: 34.9 kg
	11	29	4.6	10.8	Intestinal bacterial overgrowth, commencement of TPN for weight loss between months 7 and 10. Weight: 35 kg
D	Pre-therapy (28 years)	-	21.0	31.0	Sensorimotor polyneuropathy, external ophthalmoplegia, minimal intestinal dysmotility. MRC sum score: 56, sensory sum score: 21. Creatine kinase:1200 U/L. Weight 57.4 kg
	4	29	0.0	0.0	Improved distal sensation. Creatine kinase: 448 U/L. Weight: 59 kg
	18	47	0.0	0.0	MRC sum score: 74, sensory sum score: 19. Creatine kinase:406 U/L. Weight 63.2 kg
	58	47	0.0	0.2	MRC sum score: 74, sensory sum score: 19. Creatine kinase: 272 U/L. Weight 59 kg

**Table 7 ijms-22-03681-t007:** KEGG pathway analysis of miRNA target genes.

Pathway	Target Genes	*p*-Value	FDR *p*-Value
Notch signaling pathway	*APH1A*, *DLL1*, *JAG1*, *NOTCH1*, *NOTCH2*, *NUMBL*	1.1 × 10^−5^	0.0024
Adherens junction	*LEF1*, *MET*, *NECTIN1*, *PTPRM*, *SMAD4*, *WASF1*	0.0002	0.0143
p53 signalling pathway	*CCNE2*, *CDK6*, *EI24*, *IGFBP3*, *MDM4*, *SERPINE1*	0.0001	0.0143
Pancreatic cancer	*CDK6*, *E2F3*, *PGF*, *RALGDS*, *SMAD4*	0.0010	0.0460
Glycosaminoglycan biosynthesis—heparan sulfate/heparin	*GLCE*, *NDST1*, *XYLT1*	0.0008	0.0460
N-Glycan biosynthesis	*B4GALT2*, *FUT8*, *MGAT4A*, *MGAT5B*	0.0012	0.0473

## Data Availability

The raw data are available from the corresponding author on reasonable request.

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
