# Peer review of "Circulating miRNAs as Biomarkers for Mitochondrial Neuro-Gastrointestinal Encephalomyopathy"

_ijms, 2021, doi:10.3390/ijms22073681_

Round 1
Reviewer 1 Report
The present study is the first large and comprehensive investigation on the issue of circulating miRNAs as biomarkers of MNGIE. The data obtained by the authors make an excellent contribution to the rapidly developing area of c-miRNAs research in the diagnostics of human rare diseases. The design of the study is built correctly, providing a step-by-step discovery of a new insights on association of c-miRNAs with MNGIE. Authors thoroughly assess the impact of pre-analytical factors on the data obtained. The choose of the NGS methods (QiaSeq/Illumina NextSeq) is probably the best currently known solution for circulating small RNA studies. All PCR analyses were performed in strict compliance with the MIQE guidelines, which makes them highly valuable. The appropriate controls were included in each stage of the study. The advantage of the manuscript is that the authors provide concise but detailed descriptions of all methods. The bioinformatical and statistical analysis of the data is conducted at a high level and described well in the manuscript. Some aspects revealed during the study indicate the overall correctness of the obtained results. For example, the authors concluded that miR-30e-5p is the most stable miRNA between serum and plasma, which is in accordance to the previously published data obtained by NanoString method [1]. Also, authors showed that some miRNAs failed to prove the observations made by NGS in a subsequent Validation phase in serum samples compared to plasma samples. Indeed, plasma and serum have different miRNA profiles: serum has reduced total miRNA yield and diversity. MiRNAs packed in microvesicles or bound with proteins are partly caught by the clot, and some miRNA species are possibly degraded during the clotting with the release of proteases which degrade protein-miRNA complexes [2]. It was fortunate for the authors that they managed to obtain good quality libraries for sequencing using serum for the microRNA isolation. However, the final results of the study, indicating that only a three serum miRNAs have altered levels in MNGIE patients of the Validation phase, suggest that the analysis of c-miRNAs in plasma would be more informative.
Manuscript is well-written and brings valuable data worth to be published, however some issues should be clarified and discussed, some items should be added, and some corrections to the article should be made:
1) Did the authors assess the small RNA quality and concentrations before constructing an NGS libraries? In the Table 1A (line 1075), the title starts with “Sample RNA concentration and quality control evaluation..”, while section 4.4.3 states that “The concentration of the library was then determined in 2µL using a Qubit Fluorimeter (Thermo Fisher Scientific) with the dsDNA HS Assay Kit (Table A1)”. I suppose, in the title of Table A1 it should be “Library concentration” or “Small RNA library concentration”.
2) Were the NGS results, e.g. for some differentially expressed miRNAs, validated by qPCR on the same miRNA samples? In my opinion, it had to be done before the Validation stage. From the text of the manuscript it is unclear if the Candidate screening and Validation cohorts of the study share the same participants?
3) The authors indicate that hemolyzed samples were excluded from the study, but do not provide a clear methodology for hemolysis assessment. RBC hemolysis can be easily assessed by spectrophotometric methods based on the measurement of oxidized hemoglobin peak in serum or plasma samples, which is a widely used method for those studies where low levels of hemolysis are important, particularly for circulating microRNA studies. Has spectrophotometric evaluation of hemolysis been performed in Discovery, Candidate screening or Validation phases?
4) In section 2.1, the number of participants and mean age of the control groups are not given. Was the composition (group size and male/female ratio) the same as in the groups with MNGIE? If so, this should be mentioned in section 2.1 and mean age ± SD should be given not only for the MNGIE groups but also for the control groups.
Also, regarding group size, Figure 7 should indicate the number of participants in each group, because the PCA plot points overlap and it is not clear what the actual group size is.
5) In Table 2, what does TMM per group mean? (MNGIE TMM / Healthy control TMM). Is this mean TMM values in the group? If yes, it should be indicated in the head of a table or in the first line (MNGIE TMM / Healthy control TMM). Group sizes for MNGIE patients and healthy controls for each age range group should also be indicated in the Table 2.
6) In a Candidate cohort phase, the 20-29 age group with MNGIE clearly demonstrates overexpression of the hemolysis-related miRNAs (miR-451a: 3.25-fold, miR-16-5p: 2,4-fold; miR-486-5p: 2,55-fold; miR-15a-5p: 3.12-fold, miR-15b-5p: 2-fold). The increased circulating levels of these miRNAs has been previously shown to be almost linearly associated with the level of RBC hemolysis [3]. This means with a high probability that MNGIE 20-29 age group had several samples with higher level of hemolysis, or that all samples in this group had higher level of hemolysis compared to healthy controls. This fact should be mentioned and discussed in the “Discussion” section. To prove or reject this hypothesis, the spectrophotometically based hemolysis indices and/or miR-451a/miR-23a-3p ratios for 20-29 age group MNGIE patients and Healthy controls could be provided in the article.
Interestingly, no overexpression of these miRNAs were found in MNGIE patients from another age groups.
7) There is one explicit result of the article, which however does not seem to be discussed in the paper. Liver-specific miR-122-5p, which has been associated by a large number of studies with liver injury and diseases with impaired liver function [4], in this study was up-regulated in MNGIE patients in the screening and validation phases. MiR-122-5p was the most abundant one among miRNAs up-regulated in MNGIE patients >30 years based on NGS results. Authors showed that miR-122-5p is the dominant source of variation along PC2, a PC along which healthy and disease samples are visually segregated. It is very likely that the cause of this liver-specific microRNA increase in MNGIE patients is liver pathology. It is known that episodes of frank intestinal pseudo-obstruction may occur in MNGIE and some patients develop a hepatopathy with liver steatosis and cirrhosis.
Along with miR-122-5p, levels of two other miRs – miR-192-5p and miR-34a-5p – have closely similar distributions in the age groups of MNGIE patients and healthy controls (it immediately catches the eye in Figure 10). Interestingly, both these miRNAs have previously been shown to be associated with liver injury in various studies (reviewed in [4]). Circulating level of miR-34a-5p is elevated in the presence of non-alcoholic fatty liver disease (NAFLD) in children and adolescents with obesity and was proposed as a useful diagnostic biomarker of NAFLD [5].
It has been reported that miR-192-5p is abundantly expressed in hepatic tissues (the 2nd most abundant miRNA in the liver after miR-122-5p [6]), and is regarded as a potential biomarker for various hepatic disorders (reviewed in [7]). miR-192-5p plays an important role in lipid metabolism and inflammation in NAFLD, making it a promising diagnostic target for NAFLD. Recent studies have shown that serum levels of miR-192-5p were elevated in acute liver injury, positively correlated with the AST/ALT ratio and the levels of miR-122, a potent marker of liver injury and hepatic cell death.
Summing up these facts, the authors should check and discuss whether the liver pathology in MNGIE patients included in the study was the possible reason for overexpression of some miRs in serum and plasma. It would be useful to show how biochemical markers of hepatic pathology (if measured) correlated with miR-122-5p, miR-192-5p and miR-34a-5p serum/plasma levels in MNGIE patients. This side of the analysis should definitely be included in the “Discussion” section, as it sheds light on the reasons for the changes in circulating microRNA profiles in MNGIE in general.
8) In line 927: “a 75 bp read length (up to 46bp insert + 19bp 3’ linker + 10 UMIs)” it should be cleared if 10 UMIs is a number of UMIs or the length of the UMI. If it is a length of the UMI, “+ 10 bp UMI” should be used.
9) Have you evaluated the influence of a biochemical markers or clinical measurements that reflects the stage and severity of MNGIE disease and its comorbidities, on a circulating miRNA profiles in the Candidate screening and Validation cohorts?
10) In a recent study [8] it has been shown that increased variation in UMIs are required when sequencing libraries with limited complexity, such as small RNAs. The employment of UMIs to reduce PCR bias will inadvertently cause the under-estimation of expression of the more highly abundant miRNAs, thus creating the false impression that there are relatively similar levels of expression between many miRNAs whereas in reality, the few most abundant miRNAs represent the majority of total miRNA expression. Given that only 8% of total reads left after the UMI correction in your study, how can you evaluate this recent findings according to your data?
Finally, the text should be checked for correctness of the citation use, grammar, punctuation and abbreviation use, for example:
- line 794: the words “which are” are doubled
- lines 1085-1086: “MNGIE patient samples (SP, left panel) healthy control samples SC, (right panel)” should be changed to “MNGIE patient samples (SP, left panel) and healthy control samples (SC, right panel)”
- line 385: the words “to the” are doubled
- lines 507-514: please check the punctuation – some commas and parenthesis are missing
- line 554: what does the [x] citation mean? should be corrected
- line 645/649: you cite [37] which is stated as Wang et al. but in reference list [37] refers to Yong et al.
- lines 645-649: “Fan and coworkers ... [37]” but [37] is Yong et al. and the article by Fan et al (which is, I suggest, “Fan F, Zhuang J, Zhou P, Liu X, Luo Y. MicroRNA-34a promotes mitochondrial dysfunction-induced apoptosis in human lens epithelial cells by targeting Notch2. Oncotarget. 2017”) do not appear in the “References” section
- regular lack of a space between the text and citation, e.g. “vesicles[23]”
- line 947: extra parenthesis
- line 390: “serum seq” should be changed to “serum miRNA-seq”
- line 165: parenthesis is needed
References
- Foye C, Yan IK, David W, Shukla N, Habboush Y, Chase L, Ryland K, Kesari V, Patel T. Comparison of miRNA quantitation by Nanostring in serum and plasma samples. PLoS One. 2017 Dec 6;12(12):e0189165.
- Dufourd T, Robil N, Mallet D, Carcenac C, Boulet S, Brishoual S, Rabois E, Houeto JL, de la Grange P, Carnicella S. Plasma or serum? A qualitative study on rodents and humans using high-throughput microRNA sequencing for circulating biomarkers. Biology Methods and Protocols. 2019;4(1):bpz006.
- Kirschner MB, Edelman JJ, Kao SC, Vallely MP, Van Zandwijk N, Reid G. The impact of hemolysis on cell-free microRNA biomarkers. Frontiers in genetics. 2013 May 24;4:94.
- Lin H, Ewing LE, Koturbash I, Gurley BJ, Miousse IR. MicroRNAs as biomarkers for liver injury: Current knowledge, challenges and future prospects. Food and Chemical Toxicology. 2017 Dec 1;110:229-39.
- Oses M, Margareto Sanchez J, Portillo MP, Aguilera CM, Labayen I. Circulating miRNAs as biomarkers of obesity and obesity-associated comorbidities in children and adolescents: a systematic review. Nutrients. 2019 Dec;11(12):2890.
- Gu Y, Wei X, Sun Y, Gao H, Zheng X, Wong LL, Jin L, Liu N, Hernandez B, Peplowska K, Zhao X. miR-192-5p silencing by genetic aberrations is a key event in hepatocellular carcinomas with cancer stem cell features. Cancer research. 2019 Mar 1;79(5):941-53.
- Ren FJ, Yao Y, Cai XY, Fang GY. Emerging Role of MiR-192-5p in Human Diseases. Frontiers in Pharmacology. 2021 Feb 23;12:160.
- Saunders K, Bert AG, Dredge BK, Toubia J, Gregory PA, Pillman KA, Goodall GJ, Bracken CP. Insufficiently complex unique-molecular identifiers (UMIs) distort small RNA sequencing. Scientific reports. 2020 Sep 3;10(1):1-9.
A PDF version of the review is in the attachment.

Author Response
The authors thank the reviewer for their constructive and helpful feedback and provide the following to address the points raised.
1. Did the authors assess the small RNA quality and concentrations before constructing an NGS libraries? In the Table 1A (line 1075), the title starts with “Sample RNA concentration and quality control evaluation..”, while section 4.4.3 states that “The concentration of the library was then determined in 2µL using a Qubit Fluorimeter (Thermo Fisher Scientific) with the dsDNA HS Assay Kit (Table A1)”. I suppose, in the title of Table A1 it should be “Library concentration” or “Small RNA library concentration”.
- Thank you for highlighting this ambiguity. We have amended the title of Table A 1 to read: “Small RNA library concentration ……..” Line 1286
2. Were the NGS results, e.g. for some differentially expressed miRNAs, validated by qPCR on the same miRNA samples? In my opinion, it had to be done before the Validation stage. From the text of the manuscript it is unclear if the Candidate screening and Validation cohorts of the study share the same participants?
- We agree, this does require clarification in the text. We have included the following sentence: “Different participants were recruited into the Discovery, Candidate and Validation phases. “ Line 893.
3. The authors indicate that hemolyzed samples were excluded from the study, but do not provide a clear methodology for hemolysis assessment. RBC hemolysis can be easily assessed by spectrophotometric methods based on the measurement of oxidized hemoglobin peak in serum or plasma samples, which is a widely used method for those studies where low levels of hemolysis are important, particularly for circulating microRNA studies. Has spectrophotometric evaluation of hemolysis been performed in Discovery, Candidate screening or Validation phases?
- We have amended the sentence to include the haemolysis assessment methodology: “Samples were visually inspected for pink colouration and those which indicated haemolysis were excluded from the study.” Line 918
4. In section 2.1, the number of participants and mean age of the control groups are not given. Was the composition (group size and male/female ratio) the same as in the groups with MNGIE? If so, this should be mentioned in section 2.1 and mean age ± SD should be given not only for the MNGIE groups but also for the control groups.
- To clarify this we have added the following sentence: “Age and sex matched healthy controls were recruited into each study cohort.” Line 150
Also, regarding group size, Figure 7 should indicate the number of participants in each group, because the PCA plot points overlap and it is not clear what the actual group size is.
- We have included participant numbers in the legend to Figure 7 (now Figure 5), Line 332.
5. In Table 2, what does TMM per group mean? (MNGIE TMM / Healthy control TMM). Is this mean TMM values in the group? If yes, it should be indicated in the head of a table or in the first line (MNGIE TMM / Healthy control TMM). Group sizes for MNGIE patients and healthy controls for each age range group should also be indicated in the Table 2.
- We have amended Table 2 to include the group sizes and Mean TMM. Lines 352 and 351, respectively.
6. In a Candidate cohort phase, the 20-29 age group with MNGIE clearly demonstrates overexpression of the hemolysis-related miRNAs (miR-451a: 3.25-fold, miR-16-5p: 2,4-fold; miR-486-5p: 2,55-fold; miR-15a-5p: 3.12-fold, miR-15b-5p: 2-fold). The increased circulating levels of these miRNAs has been previously shown to be almost linearly associated with the level of RBC hemolysis [3]. This means with a high probability that MNGIE 20-29 age group had several samples with higher level of hemolysis, or that all samples in this group had higher level of hemolysis compared to healthy controls. This fact should be mentioned and discussed in the “Discussion” section. To prove or reject this hypothesis, the spectrophotometically based hemolysis indices and/or miR-451a/miR-23a-3p ratios for 20-29 age group MNGIE patients and Healthy controls could be provided in the article.
Interestingly, no overexpression of these miRNAs were found in MNGIE patients from another age groups.
- To address this, we have added Figure A1 showing the expression levels of endogenous miRNAs. Line 1289.
- We have also added the following sentence to section 2.3.1: “Expression levels of miR-103a-3p, miR-191-5p, miR-451a, miR-23a-3p and miR-30c-5p were within the expected range for miRNAs in biofuilds. The dCp(miR23a-miR451a) values were 5 or less, indicating no haemolysis (Figure A2)”. Line 269
- We have added the following to the discussion: “Of note, the 20-29 age group patient group demonstrated an overexpression of the haemolysis-related miRNAs (miR-451a, miR-16-5p, miR-486-5p, miR-15a-5p and miR-15b-5p) compared to healthy controls [42]. Although results from the quality control analysis demonstrated the dCp(miR23a-miR451a) levels were higher in the patient samples (range 0.2-2.5, samples SP3, 4, 11, 14, 24, 25 and 26) compared to healthy control samples (range 0.9 to 1.1, samples SC3, 4, 11, 14, 24, 25 and 26), these were well below the value that would normally indicate haemolysis.” Line 717
- We have also included the recommended reference, reference 42.
7. There is one explicit result of the article, which however does not seem to be discussed in the paper. Liver-specific miR-122-5p, which has been associated by a large number of studies with liver injury and diseases with impaired liver function [4], in this study was up-regulated in MNGIE
- We have added the following the discussion: “One of the typical features of full-blown MNGIE is the hepatic steatosis and cirrhosis, either disease related or induced by the long-term use of total parenteral nutrition. Interestingly, all three miRNAs have previously been shown in a number of studies to be associated with liver pathology, and consequently have been proposed as useful diagnostic biomarkers of liver injury [38-41]. The overexpression pattern of these miRNAs across the three disease age groups may therefore be explained by progressing liver pathology”. Line 707.
8. In line 927: “a 75 bp read length (up to 46bp insert + 19bp 3’ linker + 10 UMIs)” it should be cleared if 10 UMIs is a number of UMIs or the length of the UMI. If it is a length of the UMI, “+ 10 bp UMI” should be used.
- We have inserted “bp”. Line 1079
9. Have you evaluated the influence of a biochemical markers or clinical measurements that reflects the stage and severity of MNGIE disease and its comorbidities, on a circulating miRNA profiles in the Candidate screening and Validation cohorts?
- No, we haven’t, but we will be utilizing this miRNA panel in our clinical trial of EE-TP where we will be measuring a number of biochemical and clinical parameters.
10. In a recent study [8] it has been shown that increased variation in UMIs are required when sequencing libraries with limited complexity, such as small RNAs. The employment of UMIs to reduce PCR bias will inadvertently cause the under-estimation of expression of the more highly abundant miRNAs, thus creating the false impression that there are relatively similar levels of expression between many miRNAs whereas in reality, the few most abundant miRNAs represent the majority of total miRNA expression. Given that only 8% of total reads left after the UMI correction in your study, how can you evaluate this recent findings according to your data
- In our study we discarded sequences that were less than 10 bp UMI which we believe addresses the findings of Saunders et al. who demonstrated that a UMI sequence in excess of eight nucleotides will be required to resolve the relative expression levels of the more abundant miRNAs.
Finally, the text should be checked for correctness of the citation use, grammar, punctuation and abbreviation use:
- Line 794: the words “which are” are doubled: corrected
- lines 1085-1086: “MNGIE patient samples (SP, left panel) healthy control samples SC, (right panel)” should be changed to “MNGIE patient samples (SP, left panel) and healthy control samples (SC, right panel)” Corrected.
- line 385: the words “to the” are doubled. Corrected
- lines 507-514: please check the punctuation – some commas and parenthesis are missing, Corrected
- line 554: what does the [x] citation mean? should be corrected. Corrected.
- line 645/649: you cite [37] which is stated as Wang et al. but in reference list [37] refers to Yong et al. Corrected
- lines 645-649: “Fan and coworkers ... [37]” but [37] is Yong et al. and the article by Fan et al (which is, I suggest, “Fan F, Zhuang J, Zhou P, Liu X, Luo Y. MicroRNA-34a promotes mitochondrial dysfunction-induced apoptosis in human lens epithelial cells by targeting Notch2. Oncotarget. 2017”) do not appear in the “References” section. We have inserted missing reference and amended numbering.
- regular lack of a space between the text and citation, e.g. “vesicles[23]” We have corrected this
- line 947: extra parenthesis. We have removed this.
- line 390: “serum seq” should be changed to “serum miRNA-seq”. We have amended this.
- line 165: parenthesis is needed. We have amended this.
Reviewer 2 Report
Great scientific work!
Well written scientific work with interesting experimental design, including four study phase and with very important and practical impact in this ultra-rare disease. In future, these results will provide for clinicians objective assessments of clinical status of MNGIE patients, stage of disease and outcome measures for assessing therapeutic intervention efficacy.
Very informative article with many detailed tables and figures.
The only comments, some figures (2, 3, 4, 9, 10, 16 and A1) are blurred and poorly readable.
Author Response
The authors thank the reviewer for their constructive and helpful feedback and provide the following to address the points raised.
The only comments, some figures (2, 3, 4, 9, 10, 16 and A1) are blurred and poorly readable.
- We have inserted clearer Figures
Reviewer 3 Report
In this paper the authors present a workflow for candidate miRNA screening in the disease MNGIE. The aim of the study has great relevance because circulating miRNAs are considered to be promising candidates in liquid biopsy. The manuscript contains interesting results that were obtained by modern molecular biological methods and supported by several statistical approaches. However, the article contains a lot of non-relevant details that masks the main conclusions. It is complicated to understand the experimental design as well as the necessity of the several steps in order to find the candidate miRNAs in the end. The manuscript is too long, 50 pages with the supplementary, and contains 20 Figures, 7 Tables and 3 additional Figures and 4 Tables in the supplementary. Most of these do not present relevant information. Due to these reasons the whole manuscript requires thorough revision and the main conclusions need to be reconsidered. Another weakness of the study is the low sample size that makes the reliability of the results questionable.
My major comments are the following:
- The introduction is too long, it should be shortened in the light of the later results. There is too much information about the disease but lacks information about the application of miRNAs in liquid biopsy.
- The results section needs to be completely rewritten. Figure 18 that presents the experimental design needs to be presented in the beginning of the results section. The link between the different steps is not clear. E.g. why the discovery cohort was required? The results of this phase were not applied in the further steps. In the second step you sequenced the miRNA content of several samples that gave more information.
- It is required to reconsider all the Figures. Those that provide information about the quality of the samples, extraction efficiency or sequencing (e.g. Figure 1, 2, 4, 5, 6, 9, 11, 12) need to be removed or relocated to the supplementary. Figure 3 is also not necessary since its information can be seen from the Fc values presented in Table 1. Please discuss the relevance of Figure 7 and 8. The information content of Figure 10 is not clear either. Are these the expression differences - same as Fc values presented in Table 2. If yes, that it needs to be removed as well. What is signature AUC?
- The authors aimed to find good reference miRNAs. But the results are not clearly presented. Both 2.3.4 and 2.4.2 sections present the candidate reference screening? These information need to be presented in the same section. Which one was used later?
- It needs to be clearly discussed what is the main conclusion of using serum vs plasma as a source for miRNA isolation. In serum you identified less miRNAs. Please reconsider the necessity of these results. It would be enough to state that plasma proved to be a better choice than serum than use the results obtained from plasma only.
- In the end it is required to clearly define how the 5 miRNAs were collected from the previous steps. Some details are presented of miRNA expression and physiological characteristics of the patients in the Performance phase. However, it is not discussed well that what the significance of these results is. How these miRNAs can be applied as diagnostic biomarkers?
- The materials and methods section is too long as well and contains non-relevant details. E.g. the protocols of isolating kits. It needs to be shortened. Figure 19 and 20 need to be deleted or removed to the supplementary.
- The discussion part needs thorough revision. It is too long and the results are repeated. However, the significance of the results is not discussed well and lacks important references about the single miRNAs as well as about their application in liquid biopsy. It is also true for the conclusion section.
- Another weakness of the study is that the sample size is extremely low that makes the reliability of the results questionable. Due to this reason it needs to be clearly stated in the discussion section that this study needs to be considered as a pilot study and the reliability of these miRNAs as biomarkers needs to be further investigated in larger sample size.
- The supplementary contains non-relevant information in this form (e.g. RNA concentrations and quality). All of these can be removed and the deleted figures from the results section might be relocated here.
Author Response
The authors thank the reviewer for their constructive and helpful feedback and provide the following to address the points raised.
My major comments are the following:
The introduction is too long, it should be shortened in the light of the later results. There is too much information about the disease but lacks information about the application of miRNAs in liquid biopsy.
- We have removed some information related to MNGIE. Lines 76 and 85
- We consider the focus of this paper is to establish with the reader the need for a biomarker of this ultra- rare disease (less than 200 patients diagnosed globally), rather than to address the technical applications of miRNAs in liquid biopsy.
- We have removed the information related to the study design and transferred this to Line 132
The results section needs to be completely rewritten. Figure 18 that presents the experimental design needs to be presented in the beginning of the results section. The link between the different steps is not clear. E.g. why the discovery cohort was required? The results of this phase were not applied in the further steps. In the second step you sequenced the miRNA content of several samples that gave more information.
- We agree that Figure 18 is better placed earlier on in the manuscript and have moved it to the results section. This is now labelled Figure 1. We believe this should now lead the reader through the methods section.
- We have also included the following to enable the reader to follow the work flow more readily: “The study was conducted in four phases according to the study design depicted in Figure 1. The aim of the discovery phase, was to permit a power calculation for sample size determination for subsequent phases. In the second, candidate screening phase, samples were analysed by next generation sequencing (NGS). The aim of this phase was to identify suitable candidate miRNAs and normalisers for verification by real-time quantitative PCR (RT-qPCR) in the validation phase. An additional aim of the validation phase was to determine whether the type of blood biofluid (plasma or serum) had an effect on the miRNA measurement. Finally, the response of the validated miRNAs to the compassionate treatment of patients with erythrocyte encapsulated thymidine phosphorylase (EE-TP), an enzyme replacement therapy under clinical development for MNGIE, was assessed in the performance phase of the study.” Line 132.
It is required to reconsider all the Figures. Those that provide information about the quality of the samples, extraction efficiency or sequencing (e.g. Figure 1, 2, 4, 5, 6, 9, 11, 12) need to be removed or relocated to the supplementary. Figure 3 is also not necessary since its information can be seen from the Fc values presented in Table 1. Please discuss the relevance of Figure 7 and 8. The information content of Figure 10 is not clear either. Are these the expression differences - same as Fc values presented in Table 2. If yes, that it needs to be removed as well. What is signature AUC?
- We agree that the Figures relating to extraction efficiency or sequencing are better placed in the Supplementary section and have relocated these. The hierarchical cluster analysis has been maintained as we feel that this enables the reader to immediately interpret the data.
- We have retained Figure 3 as this provides a visual representation of the median and quartiles which is not present in Table 1, and Table 1 provides the p
- We have added the following to explain the relevance of Figure 7 & (now Figure 5): “thus demonstrating the similarity of samples within the study groups.” Line 318
- We have added the following to explain the relevance of Figure 8 (now Figure 6): By examining the component loading it was shown that miR-122-5P separates healthy from disease samples and was therefore the primary driver of this segregation, Figure 6.” Line 335
- The data in Figure 8 was obtained after constructing a multinomial elastic-net penalised regression model of the data in Table 2 and this resulted in the refined signature of 41 miRNAs with an AUC of 0.803 (signature AUC) by receiver operating characteristic analysis, as shown in Figure 10 (now Figure 8). To make this clearer we have amended the legend in Figure 8 to the following: “Refined miRNA signature consisting of 41 miRNAs identified after conducting a multinomial elastic-net regression analysis of the 80 differentially expressed miRNAs. These miRNAs were selected based on their model coefficients not being shrunk to zero after cross-validation of the model fit”. Line 423
The authors aimed to find good reference miRNAs. But the results are not clearly presented. Both 2.3.4 and 2.4.2 sections present the candidate reference screening? These information need to be presented in the same section. Which one was used later?
- Section 2.3.4 used data generated from the candidate study to identify candidate miRNAs that would be suitable to use as normalizers in the downstream qPCR validation study. Section 2.4.2 relates to determining whether the candidate normalisers represented the most stable reference genes, or whether there were other more suitable genes. We have therefore kept these sections separate.
It needs to be clearly discussed what is the main conclusion of using serum vs plasma as a source for miRNA isolation. In serum you identified less miRNAs. Please reconsider the necessity of these results. It would be enough to state that plasma proved to be a better choice than serum than use the results obtained from plasma only.
- We consider these results to be important as different centres use different protocols, and for rare diseases obtaining sufficient material for analysis is challenging.
- We have added the following sentence to our argument for this investigation: “and the use of this biofluid in future studies is highly recommended.” Line 751
- We have amended the abstract and conclusion to recommend the use of the plasma miRNA panel. Line 52 and Line 1211
In the end it is required to clearly define how the 5 miRNAs were collected from the previous steps. Some details are presented of miRNA expression and physiological characteristics of the patients in the Performance phase. However, it is not discussed well that what the significance of these results is. How these miRNAs can be applied as diagnostic biomarkers?
- We have added a section regarding their relevance to liver disease. Line 714. In line 779 we recommend: “To robustly test the performance of this mirRNA panel and correlate miRNA expression profiles with clinical outcome data, future investigations would benefit from operating in tandem with a clinical trial of a therapeutic intervention, thereby exploiting the methodological rigor of a regulatory approved study design.”
The materials and methods section is too long as well and contains non-relevant details. E.g. the protocols of isolating kits. It needs to be shortened. Figure 19 and 20 need to be deleted or removed to the supplementary.
- We consider Figures 19 and 20 aid the understanding of the work flow and by placing them in the Supplementary section it will require the reader to refer to another section of the paper.
- We agree that the methods are long, however we have reported this study according to the guidelines for MIQE and have included information that deviates from standard kits. As we are recommending the application of this miRNA panel in regulatory approved clinical trials, it is essential that concise information is provided whereby data generated from different studies can be compared. This is particularly the case for rare diseases, where limited data is generated and where there are often no validated outcome measures available.
The discussion part needs thorough revision. It is too long and the results are repeated. However, the significance of the results is not discussed well and lacks important references about the single miRNAs as well as about their application in liquid biopsy. It is also true for the conclusion section.
- We have removed lines 661 to 664
- We have added the following to address significant of the identified miRNAs: “One of the typical features of full-blown MNGIE is the hepatic steatosis and cirrhosis, either disease related or induced by the long-term use of total parenteral nutrition. Interestingly, all three miRNAs have previously been shown in a number of studies to be associated with liver pathology, and consequently have been proposed as useful diagnostic biomarkers of liver injury [38-41]. The overexpression pattern of these miRNAs across the three disease age groups may therefore be explained by progressing liver pathology” Line 714.
- The aim of this study was to identify a biomarker panel for a rare disease, and therefore the focus was using liquid biopsy as a tool.
Another weakness of the study is that the sample size is extremely low that makes the reliability of the results questionable. Due to this reason it needs to be clearly stated in the discussion section that this study needs to be considered as a pilot study and the reliability of these miRNAs as biomarkers needs to be further investigated in larger sample size.
- We agree that the study sample size is small. MNGIE is an ultra-rare disease with less than 200 patients diagnosed globally. It is very unlikely that a larger study could be conducted of this disease. The aim of the discovery phase was to establish the sample size required to detect a difference, and we are confident that this has enabled us to obtain reliable results.
The supplementary contains non-relevant information in this form (e.g. RNA concentrations and quality). All of these can be removed and the deleted figures from the results section might be relocated here.
- This information is included in line with the essential and desirable reporting requirements of the guidelines for MIQE.
Round 2
Reviewer 1 Report
I estimate the work done by the authors to improve the article as satisfactory.
However, I still recommend that the authors pay attention to some additional points in the manuscript that can be corrected or improved before publication:
- In the PDF version of the revised manuscript, Figure A2 in the Appendix A is not displayed correctly. I cannot see it and so I cannot provide any comments on it.
- Please check the use of “dCp” and “dCq” terms in the manuscript. Simultaneous presence of the both terms is confusing, moreover it is not clear what does “dCp” mean.
Finally, here I provide my opinion on the three main limitations of this study. I assume it will be helpful to readers.
- Serum instead of plasma as an initial choice of biofluid to measure circulating miRNAs on the NGS phase. As a conclusion, the authors finally have found and stated that in future studies the use of plasma instead of serum is highly recommended.
- The lack of spectrophotometric hemolysis assessment of plasma samples. Hemolysis was assessed in plasma samples only by visual investigation, or by the estimation of dCq (miR-23a-3p and miR-451a).
- The authors do not provide a data on the influence of biochemical markers or clinical measurements that reflects the stage and severity of MNGIE disease and its comorbidities on a circulating miRNA profiles in the Candidate screening and Validation cohorts.
Author Response
In the PDF version of the revised manuscript, Figure A2 in the Appendix A is not displayed correctly. I cannot see it and so I cannot provide any comments on it.
- We are sorry that you were unable to view Figure A2”. For some reason in our version of the PDF this Figure is displayed correctly. We have reloaded this Figure and have also asked the Editor to check if they have experienced the same problem.
Please check the use of “dCp” and “dCq” terms in the manuscript. Simultaneous presence of the both terms is confusing, moreover it is not clear what does “dCp” mean.
- We have changed the use of Crossing point (Cp) which was displayed on the graph, to Cq. Lines 271 and 720
- To clarify dCq we have added : “The difference between miR23a and miR451a values, dCq (miR23a-miR451a),…….” Line 271
Finally, here I provide my opinion on the three main limitations of this study. I assume it will be helpful to readers.
- Serum instead of plasma as an initial choice of biofluid to measure circulating miRNAs on the NGS phase. As a conclusion, the authors finally have found and stated that in future studies the use of plasma instead of serum is highly recommended.
- We have added the following sentence: “Indeed, the selection of serum, rather than plasma in the candidate screening phase represents a study limitation.” Line 755
- The lack of spectrophotometric hemolysis assessment of plasma samples. Hemolysis was assessed in plasma samples only by visual investigation, or by the estimation of dCq (miR-23a-3p and miR-451a).
- We have added the following sentence: “A spectrophotometric assessment of samples for haemolysis would have complimented the assessment of the haemolysis dependent miRNAs.” Line 723
- The authors do not provide a data on the influence of biochemical markers or clinical measurements that reflects the stage and severity of MNGIE disease and its comorbidities on a circulating miRNA profiles in the Candidate screening and Validation cohorts.
- We have included the following sentence: “Data correlating biochemical markers and clinical measurements with the stage and severity of MNGIE disease and its comorbidities on the circulating miRNA profiles in the Candidate screening and Validation cohorts would have strengthened this study.” Line 782